# Gauge Flow Matching: Efficient Constrained Generative Modeling over General Convex Set and Beyond

**Xinpeng Li**[1], **Enming Liang**[1,*], **Minghua Chen**[1,2,*]
[1]Department of Data Science, City University of Hong Kong
[2]School of Data Science, The Chinese University of Hong Kong, Shenzhen

## Abstract

Generative models, particularly diffusion and flow-matching approaches, have achieved remarkable success across diverse domains, including image synthesis and robotic planning. However, a fundamental challenge persists: ensuring generated samples strictly satisfy problem-specific constraints — a crucial requirement for physics-informed problems, safety-critical applications, watermark embedding, etc. Existing approaches, such as mirror maps and reflection methods, either have limited applicable constraint sets or introduce significant computational overhead. In this paper, we develop gauge flow matching (GFM), a simple yet efficient framework for constrained generative modeling. Our GFM approach introduces a novel bijective gauge mapping to transform generation over arbitrary compact convex sets into an equivalent process over the unit ball, which allows low-complexity feasibility-ensuring operations such as reflection or projection. The generated samples are then mapped back to the original domain for output. We prove that our GFM framework guarantees strict constraint satisfaction, with low generation complexity and bounded distribution approximation errors. We further extend our GFM framework to two non-convex settings, namely, star-convex and geodesic-convex sets. Extensive experiments demonstrate that GFM outperforms existing methods in both generation speed and quality across multiple benchmarks, including synthetic data, time series, and image generation.

## 1 Introduction

Generative models have emerged as powerful tools for learning complex data distributions, achieving remarkable success in diverse applications ranging from image generation to scientific simulation. Recent advances, particularly in diffusion models and flow-matching approaches, have further pushed the boundaries of what's possible in areas such as photorealistic image synthesis, molecular design, and robotic trajectory planning (Ramesh et al., 2022; Betker et al., 2023; Sun & Yang, 2023; Chi et al., 2023; Abramson et al., 2024; Zeni et al., 2025).

However, many real-world applications necessitate generation under specific constraints. For instance, protein synthesis requires adherence to structural constraints within amino acid chains. Image generation may demand precise watermark placement or consistency with physical laws. Robotic manipulation must respect joint limits and ensure obstacle avoidance. These constraints are not merely optional considerations but fundamental requirements of their respective problem domains. Generated samples must strictly satisfy these constraints to be both meaningful and practically useful within their intended applications.

Existing approaches to constrained generative modeling face significant limitations (see Table 1). They either have limited applicable constraints (e.g., box and simplex) or lack a strict feasibility guarantee for generated samples. To date, developing an efficient framework for constrained generation with feasibility guarantees over general compact sets, convex or not, remains largely open. This work proposes Gauge Flow Matching (GFM), addressing these challenges with the following contributions:

---

*Corresponding authors: Enming Liang (enming.cityu@gmail.com); Minghua Chen (minghua@cuhk.edu.cn)

▷ In Sec. 4, we propose a *bi-Lipschitz* bijective gauge mapping, generalized from the one in (Tabas & Zhang, 2022a), to transform generation over general *compact convex* sets to an equivalent process over a unit ball, which allows low-complexity feasibility-ensuring operations such as reflection or projection.

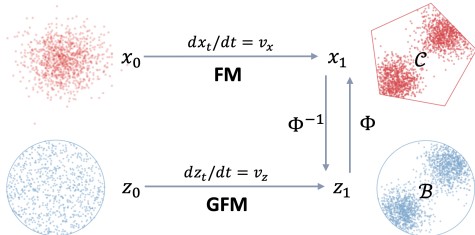

▷ In Sec. 5, we prove that the GFM framework guarantees strict constraint satisfaction and bounded distribution approximation error while incurring significantly lower computational complexity compared to SOTA constrained generative models such as regular reflection or projection-based methods.

Figure 1: Gauge flow matching framework.

▷ In Sec. 6, we further extend GFM to certain non-convex constraints, in particular *star-convex* and *geodesic-convex* sets, while inheriting the computational merits in the convex setting.

▷ In Sec. 7, we provide extensive empirical studies to demonstrate our framework's efficiency in terms of feasibility, approximation capability, and inference complexity compared to SOTA constrained generative models.

## 2 RELATED WORK

Ensuring feasibility is fundamental in constrained optimization and neural network design. While extensive work exists on constrained optimization algorithms and NN output feasibility methods (see Appendix A), constrained generative modeling presents distinct challenges. Traditional generative models like VAEs and GANs can directly apply existing NN feasibility techniques. However, diffusion and flow-based models require specialized approaches because they generate samples through forward integration with NN-approximated score functions or vector fields rather than direct NN outputs. Table 1 summarizes these specialized approaches.

**Reflected Processes** leverage reflection mechanisms to constrain trajectories within feasible regions. Methods differ in training: RDM[a] uses implicit score matching (Fishman et al., 2023), RDM[b] employs approximated denoising score matching (Lou & Ermon, 2023), and RSB utilizes iterative proportional fitting (Deng et al., 2024). RFM extends this to flow-based generation over convex sets (Xie et al., 2024). While effective, reflection-based methods incur substantial computational overhead from boundary localization and reflection calculations during forward integration. Recent work uses Metropolis sampling to reduce this burden (Fishman et al., 2024), though without strict feasibility guarantees. Additionally, these methods require the initial distribution's support to match the target set $\mathcal{C}$, introducing complexity since sampling from general convex sets is non-trivial.

**Guided Generation** incorporates auxiliary terms to enforce constraints during generation. $\Omega$-Bridge employs Doob's h-transform with time-dependent force terms (Liu & Wu, 2023), while log-barrier diffusion models enforce feasibility through barrier functions (Fishman et al., 2023). Direct projection onto the constraint set or incorporating feasible directions during forward integration for intermediate noisy samples provides another approach (Chung et al., 2022; Christopher et al., 2024; Zampini et al., 2025; Naderiparizi et al., 2025; Liang et al., 2025; Cardei et al., 2025). Rather than applying these operations to intermediate noisy samples, several methods apply them to the estimated target samples, followed by resampling or interpolation to update subsequent noisy samples (Song et al., 2023a; Cheng et al., 2024; Zhang et al., 2025; Utkarsh et al., 2025; Graikos et al., 2025; Narasimhan et al., 2025). While these methods can improve feasibility, exact projection is computationally expensive, and approximated methods lack error analysis or feasibility guarantees.

**Training/Fine-tuning** methods incorporate constraint violations as penalty terms in training objectives (Li et al., 2024) or use Lagrangian-based training with dual variable updates (Khalafi et al., 2024). Post-training fine-tuning with reward functions provides another pathway to improve constraint satisfaction (Fan & Lee, 2023; Uehara et al., 2024; Domingo-Enrich et al., 2024).

**Bijective Maps** transform complex constrained domains into unconstrained spaces or simple sets. RDM[a] maps simplexes to unit cubes for scalable score matching (Lou & Ermon, 2023). MDM uses closed-form mirror maps for simple convex sets such as balls and simplexes (Liu et al., 2024b),

Table 1: Existing study on constrained diffusion/flow-matching models over **continuous** domain.

| Method (ref. in Sec. 2) | Constraint set $\mathcal{C}$ | Initial support set[1] | Feasibility guarantee[2] | Wasserstein bound | Low generation complexity[3] |
|---|---|---|---|---|---|
| DM/FM | - | $\mathbb{R}^n$ | - | ✓ | $\mathcal{O}(\text{NFE} \cdot n^2)$ |
| RDM | Convex | $\mathcal{C}$ | ✓ | - | - |
| RSB | Smooth | $\mathcal{C}$ | ✓ | ✓ | - |
| RFM | Convex | $\mathcal{C}$ | ✓ | ✓ | - |
| MDM | Ball/Sim./Poly. | $\mathbb{R}^n$ | ✓ | - | ✓ |
| MFM | Convex | $\mathbb{R}^n$ | ✓ | ✓ | - |
| NAMM | (Non)-Convex | $\mathbb{R}^n$ | - | - | ✓ |
| Barrier-based | Convex | $\mathcal{C}$ | ✓ | - | - |
| Projection-based | (Non)-Convex | $\mathbb{R}^n$ | ✓ | - | - |
| Penalty-based | General | $\mathbb{R}^n$ | - | - | ✓ |
| **Gauge Flow Matching** | Convex | $\mathcal{B}$ | ✓ | ✓ | ✓ |

[1] The support set of initial/prior distribution matters when preparing the initial samples for training and generation, since sampling from the general convex set, even following a simple uniform distribution, is computationally expensive (Kook & Vempala, 2024).

[2] The low generation complexity of constrained generative models indicates the complexity is matched with regular DM/FM models, dominated by neural network evaluation ($\mathcal{O}(n^2)$) and scaled linearly with the number of function evaluations (NFE).

and MFM refines MDM using a modified Lipschitz mirror map (Guan et al., 2025). However, the inverse of the mirror map for a general convex domain does not admit a closed-form expression. The mapping between convex polytopes and the unit ball or unconstrained space, as applied in Diederen & Zamboni (2025), enables efficient flow matching in the transformed domain. NAMM generalizes this approach using neural networks to approximate invertible maps (Feng et al., 2024), though it lacks theoretical guarantees for feasibility and distribution approximation. Bijective mappings have also been applied to continuous embeddings of discrete categorical data (Davis et al., 2024; Williams et al., 2025). Nevertheless, constructing explicit-form bijective mappings with efficient computation for complex constraints, even general convex ones, remains non-trivial.

In summary, existing approaches either have limited applicability or lack performance guarantees. We propose a gauge mapping-based approach that, while conceptually related to mirror map methods, offers broader applicability, theoretical guarantees, and improved computational efficiency.

## 3 PROBLEM STATEMENT

We consider flow matching-based generative modeling for a data distribution $p_{\text{data}}$ over a general compact *convex* [1] set $\mathcal{C} \subset \mathbb{R}^n$. The vanilla flow-matching model (Lipman et al., 2022; Liu et al., 2022b) is trained by matching the designed conditional flow (e.g., linear flow) as:

$$\min \quad \mathcal{L}(v_\theta) = \mathbb{E}_{x_0, x_1, t}\left[\|v_\theta(x_t, t) - (x_1 - x_0)\|^2\right], \tag{1}$$

where $x_t = (1-t)x_0 + tx_1$ with $x_0 \sim p_0$, $x_1 \sim p_1$, and $t \sim \mathcal{U}((0,1))$. The minimizer of the flow matching loss in (1) yields a vector field that transforms a simple initial distribution, e.g., Gaussian $p_0 = \mathcal{N}(0, I)$, into the target data distribution $p_1 = p_{\text{data}}$ (Liu et al., 2022b). In practice, the vector field is parameterized by a neural network $v_\theta$ and optimized using samples from the target distribution according to (1). Sample generation is achieved through forward integration $x_1 = x_0 + \int_0^1 v_\theta(x_t, t)dt$, initializing from a Gaussian sample $x_0$ and following the learned vector field $v_\theta$.

**Open issue**: However, the generated samples often exhibit deviations from the constraint set $\mathcal{C}$ due to a phenomenon known as *error propagation* (Li et al., 2023c). This occurs when the approximation errors of NN-based vector fields accumulate throughout the discretized integration process, ultimately resulting in significant deviation of the generated samples from the constraint sets. Existing

---

[1]Compact convex set includes *linear-equality* and *convex-inequality* constraints. In this work, we consider the *convex-inequality* without loss of generality. For *linear-equality*, it can be embedded in an unconstrained subspace by selecting independent variables and reconstructing the dependent variables via closed-form equality solving (Donti et al., 2020; Liang et al., 2023; Ding et al., 2023), see Appendix B for details.

approaches addressing feasibility issues suffer from either limited applicability or high computational complexity (see Table 1).

## 4 GAUGE FLOW MATCHING OVER CONVEX SETS

To address these limitations of existing methods and enable efficient constrained generative modeling, we introduce our GFM framework. It employs gauge mapping—an explicit bijective mapping between two convex sets—to transform complex constrained generative modeling into an equivalent modeling over a simple unit ball. The framework **(i)** builds the flow-matching model for transformed data distribution over a unit ball through inverse gauge mapping; and **(ii)** generates samples over a unit ball via a closed-form reflection and transforms them back to the original space through forward gauge mapping.

### 4.1 GENERALIZED GAUGE MAPPING BETWEEN CONVEX SETS

We first introduce a bijective mapping between two compact convex sets in Euclidean space, known as gauge mapping (Tabas & Zhang, 2022a;b; Liang & Chen, 2025; Li et al., 2025b; Liu et al., 2025):

**Definition 4.1** (Gauge mapping). Let $\gamma_{\mathcal{C}}(x, x^\circ) = \inf\{\lambda \geq 0 \mid x \in \lambda(\mathcal{C} - x^\circ)\}$ be the *Gauge/Minkowski function* (Blanchini & Miani, 2008) given an interior point $x^\circ \in \text{int}(\mathcal{C})$. The *gauge mapping* $\Phi : \mathcal{B} \to \mathcal{C}$ can be defined between a unit $p$-norm ball and a compact convex set:

$$\Phi(z) = \frac{\|z\|_p}{\gamma_{\mathcal{C}}(z, x^\circ)} z + x^\circ, \ \forall z \in \mathcal{B}, \qquad \Phi^{-1}(x) = \frac{\gamma_{\mathcal{C}}(x - x^\circ, x^\circ)}{\|x - x^\circ\|_p}(x - x^\circ), \ \forall x \in \mathcal{C}, \quad (2)$$

As shown in Fig. 2, the gauge mapping $\Phi(\cdot)$ establishes a continuous bijective correspondence (i.e., *homeomorphism*) between any compact convex set and a unit $p$-norm ball: $\mathcal{C} = \Phi(\mathcal{B})$ and $\mathcal{B} = \Phi^{-1}(\mathcal{C})$. Intuitively, the gauge mapping transforms a unit ball into the convex set by first *translating* the unit ball to the interior point $(0 \to x^\circ)$, then *scaling* along every radial direction from this interior point such that the boundary of the ball becomes aligned with the boundary of the convex set (e.g., $z_1 \to x_1$). As a result, all level sets of $\mathcal{B}$ are mapped to level sets of $\mathcal{C}$ (e.g., $z_2 \to x_2$). The inverse mapping is similarly constructed by inverse scaling and translation back to the origin.

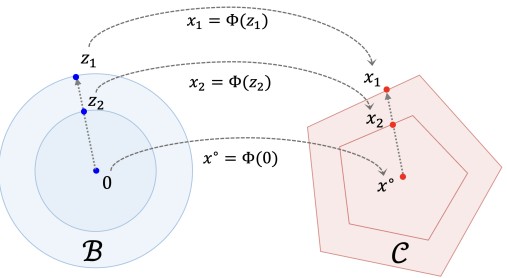

Figure 2: Gauge mapping illustration.

**Remark 1** ( Literature of Gauge function/map). Gauge/Minkowski functions $\gamma_{\mathcal{C}}$ have been extensively studied in the control and online learning communities for Lyapunov function design and efficient online optimization (Blanchini, 1995; Raković & Lazar, 2014; Mhammedi, 2022; Lu et al., 2023). While gauge mapping $\Phi$ was proposed in (Tabas & Zhang, 2022a), specifically between a polytope and a cube, we generalize it to arbitrary pairs of compact convex sets and provide efficient computation methods. Specifically, in the gauge mapping calculation, (i) gauge function $\gamma_{\mathcal{C}}(x, x^\circ)$ has *closed-form* expressions for common convex sets (e.g., linear, quadratic, and convex cones) and can be efficiently computed via *bisection* methods for general convex constraints; (ii) the interior point $x^\circ$ is obtained *once for all* by solving a convex feasibility problem offline (see details in Appendix B).

Further, we characterize the *bi-Lipschitz* property of gauge mapping, as will be demonstrated in Sec. 5, which is essential for establishing the regularity of transformed data distribution and controlling the distribution approximation error in our GFM framework.

**Proposition 4.1** (Bi-Lipschitz Property of the Gauge Mapping). *Let $\mathcal{C} \subset \mathbb{R}^n$ be a compact convex set and let $x^\circ \in \text{int}(\mathcal{C})$ be an interior point. Define the inner and outer radii with respect to $x^\circ$ as*

$$r_{\text{i}} := \sup\{r \geq 0 : \mathcal{B}_2(x^\circ, r) \subseteq \mathcal{C}\}, \quad r_{\text{o}} := \inf\{r \geq 0 : \mathcal{C} \subseteq \mathcal{B}_2(x^\circ, r)\},$$

*such that $\mathcal{B}_2(x^\circ, r_{\text{i}}) \subseteq \mathcal{C} \subseteq \mathcal{B}_2(x^\circ, r_{\text{o}})$. Then the gauge mapping $\Phi$ between $\mathcal{C}$ and 2-norm ball $\mathcal{B}_2$ satisfies the following bounds:*

$$\textit{Forward Lipschitz:} \quad \text{L}_\Phi \leq 2\,r_{\text{o}} + r_{\text{o}}^2/r_{\text{i}}, \qquad \textit{Inverse Lipschitz:} \quad \text{L}_{\Phi^{-1}} \leq 2/r_{\text{i}} \quad (3)$$

The bi-Lipschitz property of gauge mapping depends on interior point selection (Liu et al., 2025). When a near-boundary interior point is selected (i.e., $r_i \to 0$), the bi-Lipschitz constant approaches infinity, which severely distorts the data distribution and creates significant challenges for generative modeling. In practice, we aim to identify a "central" interior point (where $r_o$ is close to $r_i$) to minimize the bi-Lipschitz constant. Such an interior point can be approximately determined through constraint residual minimization (Tordesillas et al., 2023), which involves linear optimization over the target convex set and can be solved in polynomial time. While we derive explicit Lipschitz bounds for gauge mapping from a 2-norm ball to a convex set, these bi-Lipschitz bounds can be extended to arbitrary pairs of compact convex sets. For a more detailed introduction to gauge mapping and its properties, we refer readers to Appendix B.

---

**Algorithm 1** Training in GFM

**Input:** Data samples: $x_1 \sim p_{\text{data}}$ over compact convex set $\mathcal{C}$, and gauge mapping between $\mathcal{C}$ and $\mathcal{B}$.
  1: **Inverse transformation**: $z_1 = \Phi^{-1}(x_1)$
  2: Regular flow matching over $\mathcal{B}$ in Eq. (4)
**Output:** Trained NN vector field $v_\theta$.

---

**Algorithm 2** Generation in GFM

**Input:** Prior samples $z_0 \sim \mathcal{U}(\mathcal{B})$, NN vector field $v_\theta$, and gauge mapping between $\mathcal{C}$ and $\mathcal{B}$.
  1: ODE solver with **reflection** over $\mathcal{B}$ in Eq. (5)
  2: **Forward transformation**: $x_1 = \Phi(z_1)$
**Output:** Generated feasible samples $x_1$.

---

### 4.2 Training Phase of GFM

Given the gauge mapping $\Phi$ between the convex set $\mathcal{C}$ and a unit ball $\mathcal{B}$, a flow matching model is trained on the transformed space as:

$$\min \quad \mathcal{L}(v_\theta) = \mathbb{E}_{z_0, z_1, t} \left[ \| v_\theta(z_t, t) - (z_1 - z_0) \|^2 \right], \tag{4}$$

where the initial samples $z_0 \sim q_0$ are sampled as a simple prior distribution $q_0$ (e.g., uniform) supported on a unit ball $\mathcal{B}$, the terminal samples $z_1 \sim q_1$ are transformed from the samples from the original data distribution via inverse gauge mapping as $z_1 = \Phi^{-1}(x_1)$, and $z_t = (1-t)z_0 + t z_1$. In essence, GFM models the transformed data distribution as $q_{\text{data}} = \Phi_\#^{-1} p_{\text{data}}$ over a unit ball $\mathcal{B}$, where $\#$ is the push-forward operator. We then leverage the regular flow matching training approach to train a neural network vector field $v_\theta$ following (4).

**Remark 2** ( Comparison to Mirror Maps)**.** Mirror map-based generative models also employ a bijective mapping to transform constrained distributions to the unconstrained dual space (Liu et al., 2024b). However, it admits only closed-form computation for simple convex sets (e.g., balls and simplices), and it maps near-boundary samples to *infinity* in the dual spaces, challenging the transformed generative modeling both theoretically and practically. In contrast, gauge mapping is computationally efficient for any compact convex set and maintains bounded Lipschitz constants for both sides (Prop. 4.1). These properties are crucial for the regularity of transformed data distribution and efficient flow-matching training, as will be analyzed in Sec. 5.

### 4.3 Inference Phase of GFM

After training, we generate samples within the unit ball $\mathcal{B}$ following the NN vector field $v_\theta$. To constrain the generation trajectory within $\mathcal{B}$, we apply an additional reflection term (Xie et al., 2024):

$$z_1 = z_0 + \int_0^1 \left( v_\theta(z_t, t) \mathrm{d}t + \mathrm{d}\boldsymbol{L}_t \right), \tag{5}$$

where $z_0$ is sampled from a unit ball following a prior distribution (e.g., uniform), and $\boldsymbol{L}_t$ is the reflection term when $z_t$ hits the constraint boundary (Xie et al., 2024). Finally, we recover the sample to the original space following the forward gauge mapping as $x_1 = \Phi(z_1)$.

**Remark 3** ( Comparison to Regular Reflection)**.** Reflection-based mechanisms are well-established in previous works to keep the generated samples within the constraint set (Lou & Ermon, 2023; Fishman et al., 2023; Deng et al., 2024; Xie et al., 2024). However, they face several limitations: **(i)** the reflection term is computationally expensive beyond simple sets (e.g., ball and simplex); **(ii)** existing reflection generative models require a prior distribution within the target constraint set $\mathcal{C}$, which is challenging to sample from (even for uniform distribution) over general convex sets during training or generation (Kook & Vempala, 2024). These computational issues prevent reflection methods

from being applied to more complex sets. In contrast, after transforming the data distribution over a unit ball through inverse gauge mapping, we can easily sample from a unit ball (e.g., uniformly), implement a closed-form reflection term with $\mathcal{O}(n)$ algorithmic complexity, and batch computation for multiple samples, thus ensuring efficient sample generation within the ball and strict feasibility after mapping to the original data space.

# 5 PERFORMANCE ANALYSIS OF GFM

**Proposition 5.1** (Regularity of Transformed Data Distribution). *Assume that the original data distribution $p_{\text{data}}$ satisfies regularity conditions (Assumption 1 in (Wan et al., 2024)). The transformed data distribution $q_{\text{data}} = \Phi_{\#}^{-1} p_{\text{data}}$ by a bi-Lipschitz homeomorphism (e.g., gauge mapping) also satisfies those regularity conditions.*

We first assume that the original data distribution $p_{\text{data}}$ satisfies regularity conditions following (Wan et al., 2024), ensuring well-posedness of standard flow matching over the data distribution. Verifying the regularity of the transformed distribution $q_{\text{data}} = \Phi_{\#}^{-1} p_{\text{data}}$ is essential for establishing the existence and well-posedness of our gauge flow matching model. Under the bi-Lipschitz properties of gauge mapping established in Prop. 4.1, we can verify that these regularity conditions are preserved. It is worth noting that theoretical analyses of flow matching models in the literature assume various regularity conditions for data distributions (Benton et al., 2023; Gao et al., 2024a;b). We verify a general condition from recent work (Wan et al., 2024). Detailed proofs and further discussion on other common regularity conditions are provided in Appendix D.

**Proposition 5.2** (Wasserstein Bound of GFM). *Let the NN approximation error be $\epsilon_\theta^2 = \mathbb{E}_{t,z_t}\|v_\theta(z_t, t) - u(z_t, t)\|^2$, where $z_t \sim p_t$ and $p_t$ is the probability density at time $t$ driven by the target vector field $u$. Assume that $v_\theta$ is $L_\theta$-Lipschitz for $z \in \mathcal{B}$ and $t \in [0, 1]$. Denote the induced probability distribution $p_\theta^{gr}$ under $v_\theta$ with reflected generation, the Wasserstein-2 distance between the data distribution $p_{data}$ and the approximated distribution $p_\theta^{gr}$ is bounded by $\mathcal{W}_2(p_{data}, p_\theta^{gr}) \leq L_\Phi e^{1/2 + L_\theta} \epsilon_\theta$.*

The Wasserstein error of GFM is bounded by the Lipschitz of gauge mapping multiplied by the distribution error in the transformed space under reflected generation. To reduce the distribution approximation error, we can regularize the neural network Lipschitz constant $L_\theta$ or optimize the loss function such that $\epsilon_\theta$ is minimized, following standard flow matching training pipelines (Liu et al., 2022b). Specific to our GFM framework, we can further optimize the Lipschitz constant of the gauge mapping $L_\Phi$. As discussed following Prop. 4.1, we can select a "central" interior point for the gauge mapping by solving convex optimization problems, thereby reducing the bi-Lipschitz constants.

**Proposition 5.3** (Inference Complexity for GFM). *Consider a compact convex set $\mathcal{C} \subset \mathbb{R}^n$ defined by constraints $g_i(x) \leq 0$, for $i = 1, 2, \ldots, m$. The generation complexity of GFM (forward integration and gauge mapping calculation) is $\mathcal{O}(\text{NFE} \cdot n^2 + m \cdot C)$, where $C = \max\limits_{1 \leq i \leq m} \{C_i\}$ is for gauge function calculation and varies by constraint type:*

▷ *(i) For linear constraints $g_i(x) = a^\top x - b \leq 0$, $C_i = \mathcal{O}(n)$; (ii) For quadratic constraints $g_i(x) = x^\top Q x + a^\top x - b \leq 0$, $C_i = \mathcal{O}(n^2)$; (iii) For second-order cone constraints $g_i(x) = \|A^\top x + p\|_2 - (a^\top x + b) \leq 0$, $C_i = \mathcal{O}(nk)$; (iv) For matrix cone constraints $g_i(x) = \sum_{j=1}^n x_j \cdot F_j + F_0 \succeq 0$, $C_i = \mathcal{O}(nk^2 + k^3)$; where $a \in \mathbb{R}^n$, $b \in \mathbb{R}$, $Q \in \mathbb{S}_{++}^{n \times n}$, $A \in \mathbb{R}^{n \times k}$, $p \in \mathbb{R}^k$, and $F_j \in \mathbb{R}^{k \times k}$.*

▷ *For general convex function $g_i(x)$, $C_i = \mathcal{O}(c_i \log \epsilon_{\text{bis}}^{-1})$ using bisection, where $c_i$ is the complexity to evaluate $g_i(\cdot)$ given a point and $\epsilon_{\text{bis}}$ is the error tolerance in bisection.*

The forward integration complexity of our model aligns with regular flow matching approaches, requiring NFE (Number of Function Evaluations) multiplied by the evaluation complexity of $v_\theta$. The additional reflection over a unit ball incurs negligible overhead with $\mathcal{O}(n)$ complexity compared to the NN forward calculation ($\mathcal{O}(n^2)$). For gauge mapping computation, given that the interior point is pre-computed offline, (i) for common convex sets, it can be solved efficiently (Fig. 8), and (ii) in general case, the bisection algorithm achieves linear convergence with minimal per-iteration complexity, merely requiring calculating the constraint function without solving any optimization problem (Mhammedi, 2022).

## 6 GAUGE FLOW MATCHING BEYOND CONVEX SETS

We extend the gauge mapping principle to two important classes of non-convex constraint sets, broadening GFM's applicability to richer geometric settings. We include formal definitions in Appendix C and provide empirical studies in Sec. 7.

▷ **Star-convex** sets are bounded regions where the entire boundary is visible from a designated interior point (e.g., $\ell_{0.5}$-norm ball). These sets arise in robotic planning and chance-constrained optimization (Charnes & Cooper, 1959; Hansen et al., 2020; Liu et al., 2022a). The key insight is that compact star-convex sets preserve the essential properties needed for gauge function construction—both the gauge function and bijective mapping to the unit ball extend naturally from convex case (Licht, 2024).

▷ **Geodesic-convex** sets are regions on Riemannian manifolds where geodesics connecting any two points are unique and remain entirely within the set. These constraints widely arise in geometric learning applications (Chen & Lipman, 2023; Miller et al., 2024; Zaghen et al., 2025). The key insight is that geodesic-convexity ensures the exponential map at any interior point provides a local diffeomorphism (Lee, 2006), allowing gauge mapping construction in the tangent space.

## 7 EMPIRICAL STUDY

We conduct extensive simulations across illustrative examples, robotics benchmarks, and high-dimensional constrained sampling tasks to demonstrate the effectiveness of our GFM framework. Detailed experimental settings and data descriptions are provided in Appendix E.

**Baselines**: We compare against the following constrained generative models, selected for their applicability to diverse constraint types: (i) **FM**: Vanilla flow matching from Gaussian to target distribution with linear conditional flow (Lipman et al., 2022; Liu et al., 2022b); (ii) **DM**: Vanilla diffusion model with variance preserving diffusion process (Ho et al., 2020; Song et al., 2020); (iii) **Reflection**: Reflection term is applied when generated samples hit the constraint boundary (Xie et al., 2024); (iv) **Metropolis**: Metropolis sampling for approximating reflection-based generation (Fishman et al., 2024); (v) **Projection**: Orthogonal projection are applied when generated samples violate constraints (Christopher et al., 2024); (vi) **GFM**: Our framework in Sec. 4.

**Metrics**: We evaluate those baselines based on (i) constraint satisfaction (i.e., feasibility) ratio (%) of 10,000 generated samples, (ii) distribution approximation error, which is measured by Maximum Mean Discrepancy (MMD) between data samples and generated samples (Fishman et al., 2023), (iii) average per-epoch training time, including prior sampling, data transformation (in GFM only), and NN training, and (iv) inference time for generating 1,000 batched samples (unless otherwise specified, including prior sampling, forward integration, and data transformation (in GFM only).

### 7.1 SYNTHETIC EXAMPLES: CONVEX, STAR-CONVEX, AND GEODESIC-CONVEX SET

We first evaluate GFM's performance in *convex*, *star-convex*, and *geodesic-convex* domains. As shown in Figures 3 - 5, we observe: vanilla diffusion/flow-matching models fail to guarantee sample feasibility, especially when probability density concentrates near boundaries. Existing reflection, projection, and Metropolis methods face limitations with complex constraints due to: (i) expensive prior distribution sampling, *increasing both training and inference time*; (ii) constraint-specific implementation requirements incurring *longer inference time* and *limited constraint settings*.

Our GFM framework transforms generative modeling over diverse convex sets into a simpler process over a unit ball, enabling efficient prior sampling and batch-executable reflection calculations with computational efficiency comparable to vanilla FM. After generation within the unit ball, samples are mapped back to the target constraint set via low-complexity gauge mappings, ensuring strict feasibility. Further, due to the bi-Lipschitz property (Prop. 4.1) of the gauge mapping, the approximation error of GFM remains comparable to vanilla FM. This property ensures that the distortion introduced by our transformation is bounded and controlled, preserving the fidelity of the generated samples while guaranteeing their feasibility within the target constraint set.

### 7.2 CONSTRAINED CONFIGURATIONAL MODELING OF ROBOTIC ARMS

We apply GFM to robotic arm control tasks following (Fishman et al., 2023; 2024). This involves learning distributions over joint locations and manipulability ellipsoids represented by symmetric

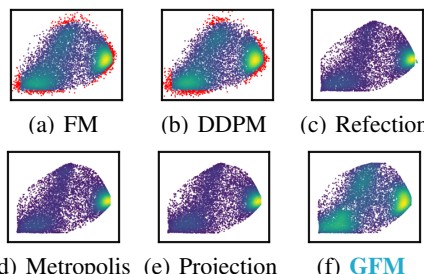

| | Feasibility (%) | MMD↓ ($\times 10^{-3}$) | Training (s) | Inference (s) |
|---|---|---|---|---|
| DDPM | 95.0 | 4.79 | 0.17 | 0.59 |
| FM | 95.9 | 8.57 | 0.18 | 0.29 |
| Refection | 100 | 25.9 | 6.40 | 14.0 |
| Metropolis | 100 | 130 | 6.40 | 6.12 |
| Projection | 100 | 93.5 | 6.40 | 7.12 |
| **GFM** | **100** | **3.50** | 0.18 | 0.63 |

(a) FM  (b) DDPM  (c) Refection

(d) Metropolis  (e) Projection  (f) **GFM**

[1] Reflection, Metropolis, and Projection models share the same NN-velocity model, and thus have the same training time per epoch.

Figure 3: Performance over joint linear and quadratic convex sets

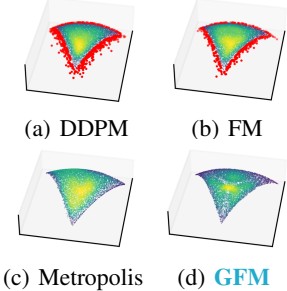

| | Feasibility (%) | MMD↓ ($\times 10^{-3}$) | Training (s) | Inference (s) |
|---|---|---|---|---|
| DDPM | 91.1 | 6.03 | 0.20 | 0.70 |
| FM | 93.4 | 4.92 | 0.22 | 0.39 |
| Reflection | 100 | 7.96 | 5.58 | 12.4 |
| Metropolis | 100 | 24.6 | 5.58 | 8.72 |
| Projection | 100 | 7.89 | 5.58 | 347 |
| **GFM** | **100** | 5.01 | 0.22 | 0.74 |

(a) DDPM  (b) FM  (c) Reflection

(d) Metropolis  (e) Projection  (f) **GFM**

[1] Sampling from a prior distribution over a non-convex domain is time-consuming for reflection and Metropolis methods.
[2] Projection onto non-convex sets incurs significant computational cost.

Figure 4: Performance over star-convex sets

| | Feasibility (%) | MMD↓ ($\times 10^{-3}$) | Training (s) | Inference (s) |
|---|---|---|---|---|
| DDPM | 93.4 | 2.84 | 0.22 | 0.73 |
| FM | 96.1 | 1.20 | 0.21 | 0.35 |
| Metropolis | 100 | 48.3 | 0.37 | 3.36 |
| **GFM** | **100** | 1.29 | 0.23 | 1.44 |

(a) DDPM  (b) FM

(c) Metropolis  (d) **GFM**

[1] Sampling from prior distributions over constrained manifolds is computationally expensive, resulting in long Metropolis inference times.
[2] High rejection rates further increase Metropolis inference time.
[3] Reflection and projection methods for constrained manifolds are not implemented due to computational issues.

Figure 5: Performance over geodesic-convex sets

positive-definite matrices with trace constraints. Figure 11 (Appendix E.6) shows generated velocity manipulation ellipsoids and trajectories. GFM successfully models this distribution while maintaining constraint satisfaction.

### 7.3 CONSTRAINED TIME SERIES GENERATION FOR TRAFFIC DATA

Time series prediction is crucial for engineering applications, yet real-world data often suffers from imbalance and scarcity due to sensor failures or privacy concerns. Synthetic data provides a promising solution for augmenting limited datasets (Narasimhan et al., 2025). We evaluate GFM on generating realistic traffic sequences from PEMS-BAY (Li et al., 2018) subject to physical (e.g., capacity bounds) and statistical (e.g., average volume) constraints that define a convex second-order cone, ensuring physically plausible and statistically consistent sequences. See details in Appendix E.8.

Table 6 shows that GFM achieves **100%** constraint satisfaction versus 88.5% for vanilla flow matching, demonstrating the necessity of constraint-aware generation. Projection baselines also achieve 100% feasibility but increase inference time from 0.31s to 49.7s (160× slower), while GFM requires only 0.43s. Critically, GFM preserves superior distribution quality with a KS statistic of 0.35 (p-value 0.42) compared to projection methods. Figure 6 confirms GFM captures characteristic traffic patterns—periodic fluctuations and sharp drops—while strictly satisfying constraints, achieving optimal balance between feasibility, fidelity, and efficiency for practical traffic data augmentation.

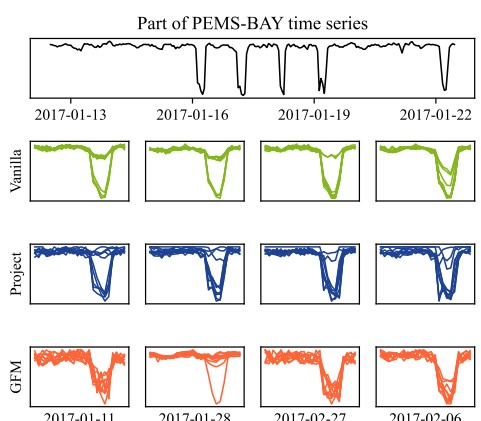

| Metric | FM | Proj. | **GFM** |
|---|---|---|---|
| Feasibility (%) | 88.5 | 100 | **100** |
| KS $\downarrow$ | 0.30 | 0.37 | 0.35 |
| p-value $\uparrow$ | 0.43 | 0.20 | 0.42 |
| Training (s) | 0.07 | 0.08 | 0.09 |
| Inference (s) | 0.31 | 49.7 | 0.43 |

[1] The Kolmogorov–Smirnov (KS) statistic measures the maximum distance between two empirical cumulative distribution functions. A KS value of 0 indicates identical distributions.

[2] The p-value is the probability of observing a KS statistic at least as large as the computed value under the identical distributions hypothesis. Larger p-values suggest stronger evidence of distributional similarity.

[3] Inference time is measured in a single generation with batch size 600.

Figure 6: Performance over constrained conditional time series generations.

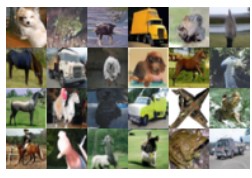 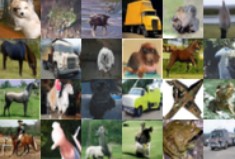 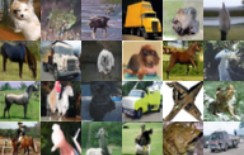 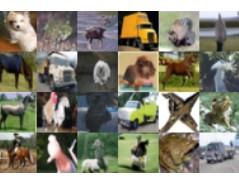

| (a) FM | (b) Projection | (c) Reflection | (d) **GFM** |
|---|---|---|---|

| **Methods** | FID (50K) $\downarrow$ | Feasibility (%) $\uparrow$ | Epoch training (s) $\downarrow$ | 5K images inference (s) $\downarrow$ | Final transformation (s) $\downarrow$ |
|---|---|---|---|---|---|
| FM | 3.57 | 76.66 | 76.7 | 155.2 | — |
| Projection | 6.88 | 100 | 100.4 | 556.6 | — |
| Reflection | 6.06 | 100 | 101.3 | 183.9 | — |
| **GFM** | 5.85 | 100 | 81.2 | 167.6 | 0.005 |

[1] We select 32 pixels as watermarks constrained via a polytope. The selected pixels in the original dataset are projected onto the polytope, while other public pixels remain unchanged in the watermarked dataset.

[2] Mirror map implementation requires an orthogonal basis polytope (Liu et al., 2024b) and is excluded for randomly sampled polytopes with general coefficients.

Figure 7: Constrained Image Generation with Embedded Watermarks

## 7.4 CONSTRAINED IMAGE GENERATION WITH EMBEDDED WATERMARKS

Watermarking generative models is critical for provenance tracking and copyright protection in AI-generated content. Following (Liu et al., 2024b), we partition pixels $\mathbf{x} \in [0, 1]^d$ into unconstrained public pixels $\mathbf{x}_1$ and watermark pixels $\mathbf{x}_2 \in \mathcal{C}$, where $\mathcal{C}$ is a random polytope for watermarking. Baselines are distilled from a publicly available checkpoint of flow matching models (Salimans & Ho, 2022; Tong et al., 2024), sharing the same U-Net architecture (34M), distilling hyperparameters (100 epochs), and sampling settings (100-step Euler method) on CIFAR-10. See details in Appendix E.9.

Fig. 7 shows that GFM maintains **100%** constraint satisfaction, achieves better FID than other constrained generative models (projection or reflection), while having similar training/inference complexity to vanilla methods. This demonstrates that GFM enables efficient, high-quality watermark embedding without compromising visual fidelity or constraint guarantees.

## 7.5 SOLUTION GENERATION FOR (RELAXED) COMBINATORIAL PROBLEMS

We further evaluate our approach on high-dimensional ($n = 10,000$) solution generation for relaxed combinatorial problems (Kook & Vempala, 2024). The target distribution follows a log-concave density and is constrained by a positive semidefinite cone and a set of linear inequalities. This target distribution encapsulates several important classes of semidefinite relaxations for classical combinatorial optimization problems (e.g, max-cut and minimum-volume covering problem). We prepare the dataset following this distribution described in (Kook & Vempala, 2024) and train our GFM model. As shown in Table 7, our method achieves **100%** feasibility rate while vanilla models fail to satisfy the constraints. GFM also maintains generation quality comparable to standard flow-matching models. Regarding inference time, the gauge mapping transformation has an explicit-form

Table 7: Solution sampling for relaxed combinatorial problems.

| Metric | $n = 10 \times 10$ | | | $n = 50 \times 50$ | | | $n = 100 \times 100$ | | |
|---|---|---|---|---|---|---|---|---|---|
| | DDPM | FM | **GFM** | DDPM | FM | **GFM** | DDPM | FM | **GFM** |
| Feasibility (%) ↑ | 47.2 | 0 | **100** | 0 | 0 | **100** | 0 | 0 | **100** |
| MMD ($\times 10^{-2}$) ↓ | 5.35 | 9.71 | 9.70 | 43.4 | 43.4 | 43.5 | 85.8 | 85.8 | 85.8 |
| Training (s) ↓ | 0.26 | 0.26 | 0.27 | 0.30 | 0.31 | 0.53 | 0.69 | 0.72 | 0.87 |
| Inference (s) ↓ | 0.97 | 0.65 | 1.17 | 2.01 | 1.66 | 11.78 | 5.93 | 5.25 | 29.12 |

computation over PSD constraints, with computational cost primarily stemming from the final-step matrix calculation (e.g., `torch.linalg.eigvalsh`) for the gauge function for $1,000$ samples in a batch. This overhead can be further reduced through the application of advanced linear algebra packages (Van de Geijn, 1997). We do not implement reflection methods since sampling the prior distribution in such a high-dimensional constraint set is computationally expensive (e.g., if we apply Ball walk, it will incur $\mathcal{O}(n^8 \log n)$ membership/evaluation complexity (Kook & Vempala, 2024)). For the projection methods, it will solve $10^5$ projection problems over the PSD cone at most ($1,000$ samples and $100$ integration steps), which is unaffordable on our computing device.

## 7.6 SCALABILITY TESTS AND ABLATION STUDY

**Scalability of gauge mapping**: We evaluate GFM's scalability in high-dimensional settings by measuring the computational cost of gauge function calculations. For closed-form gauge calculation on four common convex sets (Fig. 8), our approach maintains efficiency across dimensions up to $3,000$, demonstrating practical applicability for high-dimensional tasks. For bisection-based gauge calculation on polynomial constraints (sum-of-squares formulation, Table 12), a degree-4 polynomial with 50 variables containing $1,758,276$ monomial terms can efficiently compute gauge functions for $1,000$ samples in $0.492$ seconds. Details are provided in Appendix E.10.

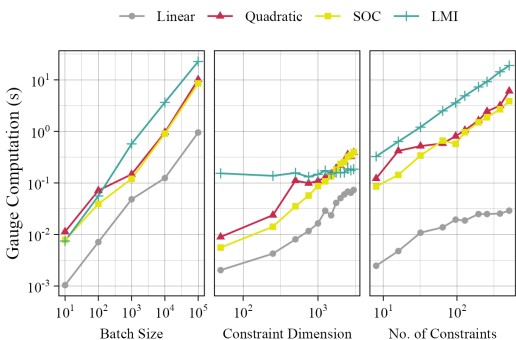

Figure 8: Gauge function computation.

**Impacts of the interior point (IP) selection**: Following Prop. 4.1, the IP selection determines the bi-Lipschitz property of gauge mapping and consequently affects generation quality. We compare *central* versus *near-boundary* IPs to demonstrate this claim, with results in Table 13.

**Comparison to Mirror Map**: We compare the Mirror Map and Gauge Map in terms of generative modeling over the simplex in 2 and 50 dimensions, with results presented in Appendix E.5.

**Impacts of generation strategies**: While we adopt *reflection* to keep samples within the unit ball (Sec. 4.3), alternatives like *projection* can also be efficiently implemented. Table 13 shows that projection is faster but leads to a larger MMD, which motivates theoretical analysis in future works.

## 8 CONCLUDING AND LIMITATIONS

We introduced Gauge Flow Matching (GFM), a framework that transforms generative modeling over arbitrary compact convex sets into an equivalent process over the unit ball through a generalized gauge mapping. This approach guarantees strict constraint satisfaction with low computational complexity and bounded distribution approximation errors, and can be extended to important non-convex settings. Our experiments demonstrate that GFM outperforms existing methods in both speed and sample quality across multiple benchmarks. Despite these advances, several limitations warrant future investigation. (i) Extending the framework to more general non-convex sets presents an important direction, which could be realized through decomposition of the non-convex set into disjoint convex or star-convex subsets, followed by conditional generative modeling over those subsets. (ii) While our primary focus is on continuous domains, extending GFM to discrete generation through relaxation or embedding techniques represents a promising avenue for future work (Davis et al., 2024). (iii) Adapting GFM to constrained one-step generation models could further improve generation efficiency beyond the current multi-step approach (Song et al., 2023b; Frans et al., 2024).

## ACKNOWLEDGMENT

This work is supported in part by a General Research Fund from Research Grants Council, Hong Kong (Project No. 11200124), a Collaborative Research Fund from Research Grants Council, Hong Kong (Project No. C1049-24G), an InnoHK initiative, The Government of the HKSAR, Laboratory for AI-Powered Financial Technologies, a Shenzhen-Hong Kong-Macau Science & Technology Project (Category C, Project No. SGDX20220530111203026), and a Start-up Research Grant from The Chinese University of Hong Kong, Shenzhen (Project No. UDF01004086). The authors would also like to thank the anonymous reviewers for their helpful comments. This work has previously been presented at the ICLR 2025 DeLTa workshop (Li et al., 2025b).

## CODE AVAILABILITY

Code is available at Github

## LLM USAGE

Large Language Models (LLMs) were used to aid in the writing and polishing of the manuscript.

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

# Contents

## A RELATED WORK ON NN FEASIBILITY

Machine learning has been increasingly integrated with optimization problems to enable efficient decision-making. Several paradigms have emerged for this integration, each addressing different aspects of the optimization process but sharing common architectural challenges. Learning to optimize trains models to iteratively refine solutions or to learn update rules that accelerate convergence (Li & Malik, 2016; Chen et al., 2022). Decision-focused learning leverages downstream optimization tasks to refine upstream prediction models (Elmachtoub & Grigas, 2022; Mandi et al., 2024; Yang et al., 2025; Guo et al., 2026). End-to-end learning trains neural networks to directly predict solutions to constrained optimization problems, serving either as warm starts for solvers or as approximate optima (Diehl, 2019; Pan et al., 2020; 2022; Li et al., 2025a; 2026). Generative modeling approaches learn solution distributions for hard optimization problems (Sun & Yang, 2023; Liang & Chen, 2024; Li et al., 2023b).

A core challenge across these paradigms is ensuring that neural network outputs satisfy the feasibility constraints of the underlying problems. Research on ensuring neural network feasibility can be categorized into several approaches:

**Basic Constraint Handling.** Specialized activation functions (e.g., Sigmoid, Softmax) address basic constraints such as box or simplex constraints (Pan et al., 2022; Donti et al., 2020). Penalties for output constraint violations can be incorporated into loss functions to improve feasibility (Cheng et al., 2019; Pan et al., 2020; Nellikkath & Chatzivasileiadis, 2021).

**Strict Satisfaction Methods.** For exact equality constraint satisfaction, prediction-then-reconstruction or completion techniques can be applied (Donti et al., 2020; Pan et al., 2022; Liang et al., 2023). For more general constraint enforcement, orthogonal or L2 projection is often employed. However, solving the projection problem, either via an iterative solver or through equivalent optimization layers (Amos & Kolter, 2017; Agrawal et al., 2019; Chen et al., 2021; Wang et al., 2023), is computationally intensive for real-time applications. More efficient bisection-based projection methods can be applied at the cost of minor optimality loss (Liang et al., 2023; 2024; Liang & Chen, 2025; Zhou et al., 2025).

**Set Parameterization Approach.** To guarantee feasibility, an inner approximation of the original constraint set can be constructed. For linear constraints, vertex networks employ convex combinations of vertices and rays to ensure policy feasibility (Frerix et al., 2020; Zheng et al., 2021). For general compact but fixed constraint sets, the probabilistic transformer utilizes feasible samples to ensure feasibility (Kratsios et al., 2021). However, scalability remains a challenge due to the exponential growth in required samples as problem dimensionality increases.

**Gauge Functions.** These methods utilize gauge functions (Blanchini & Miani, 2008) to constrain neural network outputs. A closed-form bijection, known as gauge mapping, between a hypercube and a polytope is used to bound the network output within the polytope (Tabas & Zhang, 2022a;b; Li et al., 2023a). Gauge mapping can be generalized to arbitrary convex sets (Li et al., 2025b; Liu et al., 2025). For fixed convex constraints, several works apply gauge functions to find feasible boundary solutions via scaling operations (Tordesillas et al., 2023; Konstantinov & Utkin, 2023; Li & Mohammadi, 2023).

These approaches are typically designed for the end-to-end neural network structure, which can not be directly applied to the diffusion or flow-matching based generative model due to the forward integration calculation.

## B GAUGE MAPPING OVER GENERAL CONVEX SETS

A general compact convex set encompasses both linear equality and convex inequality constraints.

$$\mathcal{C} = \{x \in \mathbb{R}^n \mid Ax = b, \ g_1(x) \leq 0, \cdots, g_m(x) \leq 0\}, \tag{6}$$

where $A \in \mathbb{R}^{r \times n}, b \in \mathbb{R}^r$, and $g_1(x), \ldots, g_m(x)$ are convex functions.

This section presents a systematic approach to handling such sets by first eliminating linear equality constraints, followed by computing gauge mappings for the remaining inequality constraints.

## B.1 Handling Linear Equality Constraints

Without loss of generality, assuming $\text{rank}(A) = r$, we partition the decision variable $x$ into $x_1 \in \mathbb{R}^{n-r}$ and $x_2 \in \mathbb{R}^r$. Accordingly, we partition matrix $A$ into $A = [A_1, A_2]$, where $A_1 \in \mathbb{R}^{r \times (n-r)}$ and $A_2 \in \mathbb{R}^{r \times r}$. The equality constraint $Ax = b$ can then be written as:

$$A_1 x_1 + A_2 x_2 = b \tag{7}$$

By choosing the partition such that $A_2$ has full rank $r$, we can express $x_2$ explicitly in terms of $x_1$:

$$x_2 = \phi(x_1) = A_2^{-1}(b - A_1 x_1) \tag{8}$$

This transformation reduces the original set to one with only inequality constraints:

$$\mathcal{C}^s = \{x_1 \in \mathbb{R}^{n-r} \mid g([x_1, \phi(x_1)]) \leq 0\} \tag{9}$$

Therefore, we only consider the inequality constraints in the main body of this work.

## B.2 Gauge Mapping for Inequality Constraints

Without loss of generality, we consider $\mathcal{C} = \{x \mid g_1(x) \leq 0, \ldots, g_m(x) \leq 0\}$.

We define the following metrics for a compact convex set.

**Definition B.1** (Point-to-boundary distance and its inverse (Tordesillas et al., 2023)). Let $\mathcal{C} \subset \mathbb{R}^n$ be a compact convex set and $x^\circ \in \text{int}(\mathcal{C})$ an interior point. For any unit vector $v \in \mathbb{S}^{n-1} = \{u \in \mathbb{R}^n \mid \|u\| = 1\}$, we define the interior-point-to-boundary distance function $d_\mathcal{C} : \text{int}(\mathcal{C}) \times \mathbb{S}^{n-1} \to \mathbb{R}_+$ along direction $v$ as

$$d_\mathcal{C}(x^\circ, v) = \sup\{\lambda \geq 0 \mid x^\circ + \lambda v \in \mathcal{C}\}. \tag{10}$$

The inverse distance function $\kappa_\mathcal{C} : \text{int}(\mathcal{C}) \times \mathbb{S}^{n-1} \to \mathbb{R}_+$ is defined as $\kappa_\mathcal{C}(x^\circ, v) := 1/d_\mathcal{C}(x^\circ, v)$.

For a compact convex set, the minimum and maximum interior point-to-boundary distances are bounded and are defined as:

- the minimum point-to-boundary distance: $r_\text{i} = \arg\sup_{r \geq 0}\{\mathcal{B}_2(x^\circ, r) \subseteq \mathcal{C}\}$.

- the maximum point-to-boundary distance: $r_\text{o} = \arg\inf_{r \geq 0}\{\mathcal{C} \subseteq \mathcal{B}_2(x^\circ, r)\}$.

- thus, we have $\mathcal{B}_2(x^\circ, r_\text{i}) \subseteq \mathcal{C} \subseteq \mathcal{B}_2(x^\circ, r_\text{o})$

### B.2.1 Gauge Function and Properties

We then introduce the gauge function:

**Definition B.2** (Gauge/Minkowski function (Blanchini & Miani, 2008)). Let $\mathcal{C} \subset \mathbb{R}^n$ be a compact convex set with a non-empty interior. The Gauge/Minkowski function $\gamma_\mathcal{C} : \mathbb{R}^n \times \text{int}(\mathcal{C}) \to \mathbb{R}_+$ is defined as

$$\gamma_\mathcal{C}(x, x^\circ) = \inf\{\lambda \geq 0 \mid x \in \lambda(\mathcal{C} - x^\circ)\}, \tag{11}$$

where $x^\circ \in \text{int}(\mathcal{C})$ is an interior point of $\mathcal{C}$.

The Gauge function generalizes the concept of a norm. For a set $\mathcal{C}$ that is symmetric about the origin, the gauge function $\gamma_\mathcal{C}(x, 0)$ defines a norm. In particular, when $\mathcal{C} = \mathcal{B}_p = \{x \in \mathbb{R}^n \mid \|x\|_p \leq 1\}$ is the unit ball of the $p$-norm, we have $\gamma_{\mathcal{B}_p}(x, 0) = \|x\|_p$. More generally, the gauge function satisfies the following properties for all $x, y \in \mathbb{R}^n$ and $\alpha \geq 0$:

**Lemma 1** (Properties of gauge function). *The gauge function $\gamma_\mathcal{C}(x, x^\circ)$ with respect to a compact convex set $\mathcal{C}$ and an interior point $x^\circ \in \text{int}(\mathcal{C})$ satisfies the following properties:*

- *Non-negativity: $\gamma_\mathcal{C}(x, x^\circ) \geq 0$*

- *Positive homogeneity: $\gamma_\mathcal{C}(\alpha x, x^\circ) = \alpha \gamma_\mathcal{C}(x, x^\circ)$ for $\alpha \geq 0$.*

- *Subadditivity: $\gamma_\mathcal{C}(x + y, x^\circ) \leq \gamma_\mathcal{C}(x, x^\circ) + \gamma_\mathcal{C}(y, x^\circ)$.*

- *Convexity: induced by positive homogeneity and subadditivity.*

- *Differentiability: under convexity, the gauge function is twice differentiable almost everywhere by Alexandrov's theorem (Rockafellar, 1999).*

- *Equivalent formulations based on (inverse) distance function:*

$$\gamma_{\mathcal{C}}(x, x^\circ) = \kappa_{\mathcal{C}}(x^\circ, x/\|x\|) \cdot \|x\| = \frac{\|x\|}{d_{\mathcal{C}}(x^\circ, x/\|x\|)} \tag{12}$$

$$\gamma_{\mathcal{C}}(x/\|x\|, x^\circ) = \kappa_{\mathcal{C}}(x^\circ, x/\|x\|) = \frac{1}{d_{\mathcal{C}}(x^\circ, x/\|x\|)} \tag{13}$$

- *Upper/lower bounds: the gauge function is bounded as: $\gamma_{\mathcal{C}}(x, x^\circ) \in [\|x\|/r_\mathrm{o}, \|x\|/r_\mathrm{i}]$*

- *Lipschitz: the gauge function has Lipschitz constant as $\frac{\|\gamma_{\mathcal{C}}(x,x^\circ) - \gamma_{\mathcal{C}}(y,x^\circ)}{\|x-y\|}\| \le \frac{1}{r_\mathrm{i}}$*

*Proof.* The non-negativity, positive homogeneity, and subadditivity are provided in (Blanchini & Miani, 2008). Convexity and differentiability are naturally implied by those properties. The equivalent formulation is derived from their definitions.

The upper and lower bounds are derived as:

$$\gamma_{\mathcal{C}}(x, x^\circ) = \frac{x}{d_{\mathcal{C}}(x^\circ, x/\|x\|)} \in [\|x\|/r_\mathrm{o}, \|x\|/r_\mathrm{i}] \tag{14}$$

The Lipschitz constant of the gauge function is derived as:

$$\gamma_{\mathcal{C}}(x, x^\circ) \le \gamma_{\mathcal{C}}(x - y, x^\circ) + \gamma_{\mathcal{C}}(y, x^\circ) \tag{15}$$

$$\gamma_{\mathcal{C}}(x, x^\circ) - \gamma_{\mathcal{C}}(y, x^\circ) \le \gamma_{\mathcal{C}}(x - y, x^\circ) \le \|x - y\|/r_\mathrm{i} \tag{16}$$

where the inequality (15) is derived by the positive homogeneity. Similarly, we have:

$$\gamma_{\mathcal{C}}(y, x^\circ) - \gamma_{\mathcal{C}}(x, x^\circ) \le \gamma_{\mathcal{C}}(y - x, x^\circ) \le \|x - y\|/r_\mathrm{i} \tag{17}$$

Combining two inequalities, we have the Lipschitz as:

$$\frac{\|\gamma_{\mathcal{C}}(x, x^\circ) - \gamma_{\mathcal{C}}(y, x^\circ)}{\|x - y\|}\| \le \frac{1}{r_\mathrm{i}} \tag{18}$$

$\square$

### B.2.2 GAUGE MAPPING AND PROPERTIES

Based on the gauge function, we can construct a bijective mapping between any pair of compact convex sets as:

**Definition B.3** (Gauge Mapping). Let $\mathcal{Z}, \mathcal{X} \subset \mathbb{R}^n$ be compact convex sets with interior points $z^\circ \in \mathrm{int}(\mathcal{Z})$ and $x^\circ \in \mathrm{int}(\mathcal{X})$, respectively.

The gauge mapping $\Phi : \mathcal{Z} \to \mathcal{X}$ is defined as:

$$\Phi(z) = \frac{\gamma_{\mathcal{Z}}(z - z^\circ, z^\circ)}{\gamma_{\mathcal{X}}(z - z^\circ, x^\circ)}(z - z^\circ) + x^\circ, \; z \in \mathcal{Z} \tag{19}$$

The inverse mapping $\Phi^{-1} : \mathcal{X} \to \mathcal{Z}$ is given by:

$$\Phi^{-1}(x) = \frac{\gamma_{\mathcal{X}}(x - x^\circ, x^\circ)}{\gamma_{\mathcal{Z}}(x - x^\circ, z^\circ)}(x - x^\circ) + z^\circ, \; x \in \mathcal{X} \tag{20}$$

In essence, the gauge mapping scales the boundary of a convex set from an interior point to another convex set and translates it to its interior point. When $\mathcal{Z}$ is a unit $p$-norm ball, the gauge mapping is simplified in Def. 4.1 as:

$$\Phi(z) = \frac{\|z\|_p}{\gamma_{\mathcal{C}}(z, x^\circ)}z + x^\circ, \; \forall z \in \mathcal{B}, \quad \Phi^{-1}(x) = \frac{\gamma_{\mathcal{C}}(x - x^\circ, x^\circ)}{\|x - x^\circ\|_p}(x - x^\circ), \; \forall x \in \mathcal{C}, \tag{21}$$

Further, the gauge mapping between $\mathcal{B}_2$ and $\mathcal{C}$ can be simplified as:

$$\Phi(z) = d_{\mathcal{C}}(x^\circ, z/\|z\|) \cdot z + x^\circ, \; \forall z \in \mathcal{B}, \quad \Phi^{-1}(x) = \frac{x - x^\circ}{d_{\mathcal{C}}(x^\circ, x - x^\circ/\|x - x^\circ\|)}, \; \forall x \in \mathcal{C} \tag{22}$$

**Proposition B.1** (Properties of gauge mapping). *The gauge mapping between any pair of compact convex sets satisfies the following properties:*

- *it is invertible.*

- *it is continuous everywhere and twice differentiable almost everywhere in both directions.*

- *it is a bi-Lipschitz homeomorphism.*

*Proof.* First, the invertibility can be easily verified by:

$$\Phi(\Phi^{-1}(x)) = \Phi\big(\frac{\gamma_{\mathcal{X}}(x - x^\circ, x^\circ)}{\gamma_{\mathcal{Z}}(x - x^\circ, z^\circ)}(x - x^\circ) + z^\circ\big) \tag{23}$$

$$= \frac{\gamma_{\mathcal{Z}}\big(\frac{\gamma_{\mathcal{X}}(x-x^\circ,x^\circ)}{\gamma_{\mathcal{Z}}(x-x^\circ,z^\circ)}(x - x^\circ), z^\circ\big)}{\gamma_{\mathcal{X}}\big(\frac{\gamma_{\mathcal{X}}(x-x^\circ,x^\circ)}{\gamma_{\mathcal{Z}}(x-x^\circ,z^\circ)}(x - x^\circ), x^\circ\big)} \frac{\gamma_{\mathcal{X}}(x - x^\circ, x^\circ)}{\gamma_{\mathcal{Z}}(x - x^\circ, z^\circ)}(x - x^\circ) + x^\circ \tag{24}$$

$$= \frac{\frac{\gamma_{\mathcal{X}}(x-x^\circ,x^\circ)}{\gamma_{\mathcal{Z}}(x-x^\circ,z^\circ)}\gamma_{\mathcal{Z}}\big((x - x^\circ), z^\circ\big)}{\frac{\gamma_{\mathcal{X}}(x-x^\circ,x^\circ)}{\gamma_{\mathcal{Z}}(x-x^\circ,z^\circ)}\gamma_{\mathcal{X}}\big((x - x^\circ), x^\circ\big)} \frac{\gamma_{\mathcal{X}}(x - x^\circ, x^\circ)}{\gamma_{\mathcal{Z}}(x - x^\circ, z^\circ)}(x - x^\circ) + x^\circ \tag{25}$$

$$= (x - x^\circ) + x^\circ = x \tag{26}$$

where Eq. (25) is derived by the positive homogeneity of the gauge function.

Second, the continuity and differentiability are derived from the properties of the gauge function and elemental compositions in the gauge mapping.

Third, the homeomorphism is derived from the continuity in both directions and the invertibility.

**Proof of Bi-Lipschitz Properties of Gauge Mapping in Prop. 4.1**

To derive the bi-Lipschitz properties, we first consider the gauge mapping between $\mathcal{B}_2$ and $\mathcal{C}$ and derive its Lipschitz constants shown in Prop. 4.1 as follows,

$$\Phi(z) = \frac{\|z\|_2}{\gamma_{\mathcal{C}}(z, x^\circ)} z + x^\circ, \ \forall z \in \mathcal{B}_2 \tag{27}$$

Differentiating $\Phi(z)$ with respect to $z$ (using the product and quotient rules) yields

$$\frac{\partial \Phi}{\partial z} = \frac{\|z\|_2}{\gamma_{\mathcal{C}}(z, x^\circ)} I + \frac{zz^\top}{\gamma_{\mathcal{C}}(z, x^\circ)\|z\|_2} - \frac{\|z\|_2}{\gamma_{\mathcal{C}}(z, x^\circ)^2} z\big(\nabla_z \gamma_{\mathcal{C}}(z, x^\circ)\big)^\top. \tag{28}$$

Taking the operator norm and applying the triangle inequality gives

$$\Big\|\frac{\partial \Phi}{\partial z}\Big\| \leq \frac{\|z\|_2}{\gamma_{\mathcal{C}}(z, x^\circ)} + \frac{\|z\|_2}{\gamma_{\mathcal{C}}(z, x^\circ)} + \frac{\|z\|_2^2}{\gamma_{\mathcal{C}}(z, x^\circ)^2} \big\|\nabla_z \gamma_{\mathcal{C}}(z, x^\circ)\big\|$$

$$\leq r_{\mathrm{o}} + r_{\mathrm{o}} + \frac{r_{\mathrm{o}}^2}{r_{\mathrm{i}}}, \tag{29}$$

where in the last inequality we have used the facts that (i) for $z \in \mathcal{B}_2$ one has $\|z\|_2 \leq 1$, (ii) the gauge function satisfies $\gamma_{\mathcal{C}}(z, x^\circ) \in [\|z\|/r_{\mathrm{o}}, \|z\|/r_{\mathrm{i}}]$, and (iii) $\|\nabla_z \gamma_{\mathcal{C}}(z, x^\circ)\|$ is bounded by $1/r_{\mathrm{i}}$. In summary, we obtain

$$\Big\|\frac{\partial \Phi}{\partial z}\Big\| \leq 2r_{\mathrm{o}} + \frac{r_{\mathrm{o}}^2}{r_{\mathrm{i}}}, \tag{30}$$

which proves that the forward Lipschitz constant of $\Phi$ satisfies

$$\text{Forward Lipschitz:} \quad \mathrm{L}_\Phi \leq 2r_{\mathrm{o}} + \frac{r_{\mathrm{o}}^2}{r_{\mathrm{i}}}. \tag{31}$$

Next, consider the inverse gauge mapping from $\mathcal{C}$ to the 2-norm ball as

$$\Phi^{-1}(x) = \frac{\gamma_{\mathcal{C}}(x - x^\circ, x^\circ)}{\|x - x^\circ\|_2}(x - x^\circ), \ \forall x \in \mathcal{C} \tag{32}$$

Differentiating with respect to $x$ gives

$$\frac{\partial \Phi^{-1}}{\partial x} = \nabla \gamma_{\mathcal{C}}(x - x^\circ, x^\circ)\left(\frac{x - x^\circ}{\|x - x^\circ\|_2}\right)^\top + \gamma_{\mathcal{C}}(x - x^\circ, x^\circ) \cdot \frac{I - \frac{(x - x^\circ)(x - x^\circ)^\top}{\|x - x^\circ\|_2^2}}{\|x - x^\circ\|_2}. \tag{33}$$

Taking norms and again using the triangle inequality leads to

$$\left\|\frac{\partial \Phi^{-1}}{\partial x}\right\| \leq \left\|\nabla \gamma_{\mathcal{C}}(x - x^\circ, x^\circ)\left(\frac{x - x^\circ}{\|x - x^\circ\|_2}\right)^\top\right\| + \gamma_{\mathcal{C}}(x - x^\circ, x^\circ)\left\|\frac{I - \frac{(x - x^\circ)(x - x^\circ)^\top}{\|x - x^\circ\|_2^2}}{\|x - x^\circ\|_2}\right\|. \tag{34}$$

Using the bound $\gamma_{\mathcal{C}}(x - x^\circ, x^\circ) \in [\|x - x^\circ\|/r_{\mathrm{o}}, \|x - x^\circ\|/r_{\mathrm{i}}]$ and the projection term has norm at most $1/\|x - x^\circ\|_2$), we have

$$\left\|\frac{\partial \Phi^{-1}}{\partial x}\right\| \leq \frac{1}{r_{\mathrm{i}}} + \frac{1}{r_{\mathrm{i}}} = \frac{2}{r_{\mathrm{i}}}. \tag{35}$$

Thus, the inverse Lipschitz constant is bounded by

$$\text{Inverse Lipschitz:} \quad \mathrm{L}_{\Phi^{-1}} \leq \frac{2}{r_{\mathrm{i}}}. \tag{36}$$

This completes the proof of Prop. 4.1.

To extend the Bi-Lipschitz properties to any pair of compact convex sets, e.g., $\mathcal{Z}$ and $\mathcal{X}$, we decompose the gauge mapping as

$$\mathcal{X} = \Phi(\mathcal{Z}) = \Phi_1(\mathcal{B}_2) = \Phi_1(\Phi_2^{-1}(\mathcal{Z})) \tag{37}$$

where $\mathcal{Z} = \Phi_1(\mathcal{B}_2)$ and $\mathcal{X} = \Phi_2(\mathcal{B}_2)$.

Leveraging the inequality of Lipschitz for mapping compositions as $\mathrm{L}_{f_1 \circ f_2} \leq \mathrm{L}_{f_1}\mathrm{L}_{f_2}$, we can conclude that the gauge mapping between any pair of convex sets is bi-Lipschitz.

$\square$

### B.2.3 COMPUTATION METHODS

**IP computation**

In practice, we seek such a "central" interior point by solving the following residual minimization problem through convex optimization in the offline phase (Tordesillas et al., 2023):

$$\min_{x^\circ} \eta \tag{38}$$

$$\text{s.t. } g_i(x^\circ) \leq \eta \quad i = 1, \cdots, m \tag{39}$$

We note that solving this convex optimization problem with a linear objective incurs only polynomial time complexity. As this computation is performed offline prior to model training, it adds negligible overhead to the overall computational cost.

**Gauge function computation**

Computation of gauge mapping essentially involves calculating the gauge function or (inverse) distance function. Following the established closed-form distance function calculation for several common convex sets in (Tordesillas et al., 2023), we give a summary in Table 8. It provides closed-form expressions for the inverse distance function across various constraint types. Most matrix calculations can be computed and stored offline before being applied for online inference.

When the inverse distance function lacks an explicit expression, we employ an efficient bisection algorithm detailed in algorithm 3. This algorithm supports batch processing, enabling efficient parallel computation for multiple inputs simultaneously.

Table 8: Closed-form Expressions for Inverse Distance Functions (Tordesillas et al., 2023)

| Constraints | Formulation | Inverse Distance Function |
|---|---|---|
| Intersections | $\{g_1(x) \leq 0, \cdots, g_m(x) \leq 0\}$ | $\kappa_g(x^\circ, v) = \max\limits_{1 \leq i \leq m} \{\kappa_{g_i}(x^\circ, v)\}$ |
| Linear | $g_L(x) = a^\top x - b \leq 0$ | $\kappa_{g_L}(x^\circ, v) = \{\frac{a^\top v}{b - a^\top x^\circ}\}^+$ |
| Quadratic | $g_Q(x) = x^\top Q x + a^\top x - b \leq 0$ | $\kappa_{g_Q}(x^\circ, v) = \{1/\text{root}(A_Q, B_Q, C_Q)\}^+$ |
| Second Order Cone | $g_S(x) = \|A^\top x + p\|_2 - (a^\top x + b) \leq 0$ | $\kappa_{g_S}(x^\circ, v) = \{1/\text{root}(A_S, B_S, C_S)\}^+$ |
| Matrix Cone | $g_M(x) = \sum_{i=1}^n x_i \cdot F_i + F_0 \succeq 0$ | $\kappa_{g_M}(x^\circ, v) = \max\{\text{eig}(L^\top(-S)L)\}^+$ |

[1] Notation: $x, a \in \mathbb{R}^n, b \in \mathbb{R}, Q \in \mathbb{S}_+^n, A \in \mathbb{R}^{n \times k}, p \in \mathbb{R}^k, F_0, \cdots, F_n \in \mathbb{R}^{k \times k}$.

[2] $A_Q = v^\top Q v$, $B_Q = 2x^{\circ\top} Q v + a^\top v$, $C_Q = x^{\circ\top} Q x^\circ + a^\top x^\circ - b$.

[3] $A_S = (A^\top v)^\top (A^\top v) - (a^\top v)^2$, $B_S = 2(A^\top x^\circ + p)^\top (A^\top v) - 2(a^\top x^\circ + b)(a^\top v)$, $C_S = (A^\top x^\circ + p)^\top (A^\top x^\circ + p) - (a^\top x^\circ + b)^2$.

[4] $H = F_0 + \sum_{i=1}^n x_i^\circ F_i$, $H^{-1} = L^\top L$, $S = \sum_{i=1}^n v_i F_i$.

[5] $(\cdot)^+ = \max(\cdot, 0)$.

[6] $\text{root}(x_1, x_2, x_3) = \frac{-x_2 \pm \sqrt{x_2^2 - 4x_1 x_3}}{2x_1}$ denotes the quadratic equation solution

[7] $\text{eig}(X) = \lambda_1, \cdots, \lambda_n$ denotes the eigenvalues satisfying $\det(X - \lambda I) = 0$. Note that only the maximum eigenvalue is needed. Thus, power iteration methods can be applied to compute it efficiently.

[8] Note that all $v$-independent terms can be computed only once and stored for use.

---

**Algorithm 3** Bisection Algorithm for Point-to-Boundary Distance

**Input:** A compact convex set $\mathcal{C}$, an interior point $x^\circ \in \text{int}(\mathcal{C})$, and a unit vector $v$.

1: Initialize: $\alpha_l = 0$ and $\alpha_u = 1$
2: **while** $|\alpha_l - \alpha_u| \geq \epsilon$ **do**
3:    **if** $x^\circ + \alpha_u \cdot v \in \mathcal{C}$ **then**
4:       increase lower bound: $\alpha_l \leftarrow \alpha_u$
5:       double upper bound: $\alpha_u \leftarrow 2 \cdot \alpha_m$
6:    **else**
7:       bisection: $\alpha_m = (\alpha_l + \alpha_u)/2$
8:       **if** $x^\circ + \alpha_m \cdot v \in \mathcal{C}$ **then**
9:          increase lower bound: $\alpha_l \leftarrow \alpha_m$
10:      **else**
11:         decrease upper bound: $\alpha_u \leftarrow \alpha_m$
12:      **end if**
13:    **end if**
14: **end while**

**Output:** $d_\mathcal{C}(x^\circ, v) \approx \alpha_m$

---

## C   GAUGE MAPPING FOR NON-CONVEX CONSTRAINTS

In this section, we extend the conventional gauge mapping over convex sets in Euclidean space to certain non-convex settings, including star-convex sets and geodesic-convex sets.

### C.1   STAR-CONVEX SET

**Definition C.1** (Star-convex Set). A set $\mathcal{S} \subset \mathbb{R}^n$ is said to be *star-convex* with respect to an interior point $x^\circ \in \text{int}(\mathcal{S})$ if for every $x \in \mathcal{S}$, the line segment connecting $x^\circ$ and $x$ lies entirely within $\mathcal{S}$, i.e., $x^\circ + \theta(x - x^\circ) \in \mathcal{S}$ for all $\theta \in [0, 1]$.

The gauge mapping framework can be extended beyond convex sets to certain non-convex domains, particularly star-convex sets. This extension significantly broadens the applicability of our GFM approach to more complex geometric constraints encountered in practical applications. A star-convex set $\mathcal{S}$ is characterized by the existence of an interior point $x^\circ$ such that any line segment connecting $x^\circ$ to any point in the set remains entirely within the set. While star-convex sets lack the full convexity property, they retain a critical radial structure that allows gauge mappings to be constructed in a similar manner.

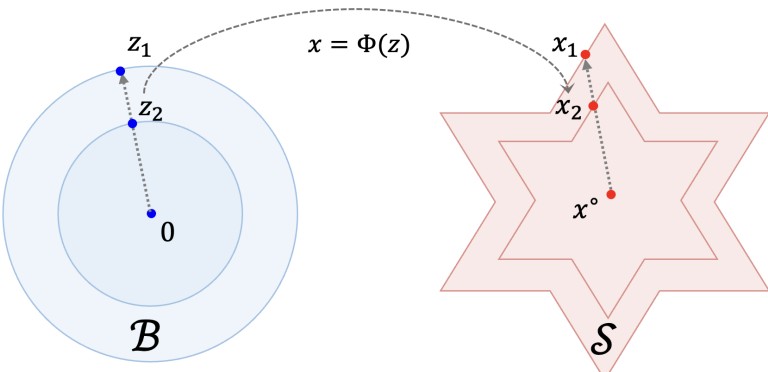

Figure 9: Gauge mapping construction for star-convex set.

For a star-convex set $\mathcal{S}$ with interior point $x^\circ$, we can define the gauge function as:

$$\gamma_{\mathcal{S}}(x, x^\circ) = \inf\{\lambda \geq 0 \mid x \in \lambda(\mathcal{S} - x^\circ)\} \tag{40}$$

This formulation captures the minimum scaling factor needed to reach point $x$ when scaling along the ray from $x^\circ$. The corresponding gauge mapping $\Phi : \mathcal{B} \to \mathcal{S}$ between a unit $p$-norm ball and the star-convex set follows the same construction as in Definition 4.1. As illustrated in Figure 9, the gauge mapping preserves the radial structure of the star-convex set while establishing a homeomorphism with the unit ball. This property is particularly valuable for handling constraints with non-convex geometries that still maintain visibility from an interior point, such as $\ell_p$-norm balls with $p < 1$ or certain non-convex polytopes arising in practical applications.

The computation of gauge functions for star-convex sets follows similar principles as for convex sets, though additional care may be needed:

- For simple star-convex sets (e.g., $\ell_p$-norm balls with $p < 1$), closed-form expressions for the gauge function can often be derived as Def. 4.1.

- For star-convex sets defined by the union of convex components or by non-convex inequalities, bisection methods remain applicable for computing the gauge function.

- Selection of the interior point $x^\circ$ becomes more critical for star-convex sets, as it determines the visibility region and thus the quality of the mapping. When multiple interior points are viable, selecting one that maximizes the minimum distance to the boundary often yields better numerical properties.

## C.2   GEODESIC-CONVEX SET

When the underlying space is a Riemannian manifold rather than a Euclidean space, the notion of straight lines is replaced by geodesics. In this context, one speaks of geodesic convexity.

**Definition C.2** (Geodesic-Convex Set). Let $(M, g)$ be a Riemannian manifold, and let $\mathcal{U} \subset M$ be a subset. $\mathcal{U}$ is *geodesically convex* if, for every pair of points $x, y \in \mathcal{U}$, there exists a unique minimizing geodesic $\eta : [0, 1] \to M$ with respect to $g$ that satisfies $\eta(0) = x, \eta(1) = y, \eta(t) \in \mathcal{U}, \forall t \in [0, 1]$.

To extend the gauge mapping on a geodesic-convex set over a manifold, we replace the linear structure with the exponential map. Since $\mathcal{U} \subset M$ is geodesic-convex, the exponential map at any interior point is a diffeomorphism onto its image (Lee, 2006). Given this property, any point $x \in \mathcal{U}$ with $x \neq x^\circ$ can be uniquely represented in *geodesic polar coordinates* as $x = \exp_{x^\circ}(r_x v_x)$, where $x^\circ \in \mathcal{U}$ is an interior point of $\mathcal{U}$, $v_x \in \mathbb{S}^{n-1} \subset T_{x^\circ} M$ is a unit vector over the tangent space $T_{x^\circ} M$, and $r_x = d_g(x, x^\circ)$ is the geodesic distance from $x^\circ$ to $x$.

Similarly, we can establish the *geodesic gauge mapping* that transforms a unit ball from the tangent space to the geodesic-convex set on the manifold. As illustrated in Figure 10, this mapping establishes a correspondence between the unit ball in the tangent space and the geodesic-convex set on the manifold. The construction respects the intrinsic geometry of the manifold while preserving the radial structure that is characteristic of gauge mappings.

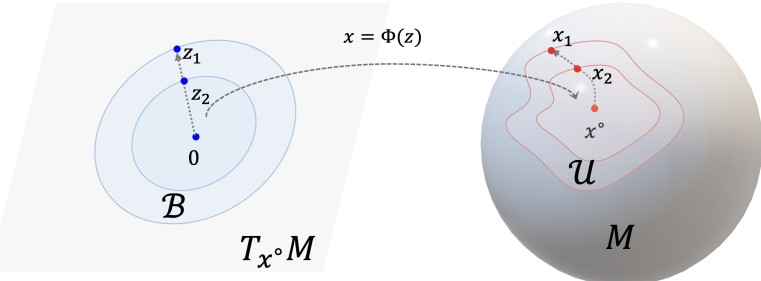

Figure 10: Gauge mapping construction for geodesic-convex set.

**Definition C.3** (Geodesic Gauge Mapping). Without loss of generality, let $\mathcal{B} \triangleq \{z \in T_{x^\circ}M \mid \|z\|_2 \leq 1\}$ denote the unit ball in the tangent space at $x^\circ$. The *geodesic gauge mapping* $\Phi : \mathcal{B} \to \mathcal{U}$ and its inverse are defined as

$$\Phi(z) = \exp_{x^\circ}(d_\mathcal{U}(x^\circ, z/\|z\|) \cdot z), \quad \forall z \in \mathcal{B} \tag{41}$$

$$\Phi^{-1}(x) = r_x/d_\mathcal{U}(x^\circ, v_x) \cdot v_x, \quad \forall x = \exp_{x^\circ}(r_x v_x) \in \mathcal{U} \tag{42}$$

where the point-to-boundary geodesic distance $d_\mathcal{U}(x^\circ, v_x)$ along $v_x$ is defined as $d_\mathcal{U}(v_x) \triangleq \sup\{\lambda \geq 0 \mid \exp_{x^\circ}(\lambda \cdot v_x) \in \mathcal{U}\}$.

Note that the point-to-boundary geodesic distance may not admit a closed-form expression; we can apply the bisection methods similar to Alg. 3 to locate the boundary points starting from an interior point.

This extension enables our GFM framework to handle constraints on manifolds, expanding its application to domains such as inverse problems with manifold constraints and atmospheric modeling over spherical geometries.

### C.3 MORE GENERAL NON-CONVEX SET

When applying our methods to more general non-convex sets, a natural choice is the class of non-convex sets that are homeomorphic to a unit ball. However, such *ball-homeomorphic* sets do not generally admit a closed-form homeomorphism. One promising approach is to apply an invertible neural network to learn the target homeomorphism, which offers both feasibility and approximation guarantees (Liang et al., 2023; 2024). For more general non-convex constraints, low-complexity constrained generation schemes remain largely unexplored, particularly for NP-hard problems where even identifying an interior point poses significant computational challenges, such as non-convex quadratic equations or mixed-integer formulations. The exploration of more general and theoretically guaranteed approaches constitutes a promising direction for future research.

## D THEORETICAL ANALYSIS OF GFM

### D.1 ASSUMPTION AND PRELIMINARY

We made the following assumptions for the error analysis, which are common for error analysis for the flow/diffusion-based generative models (Kwon et al., 2022; Benton et al., 2023; Xie et al., 2024).

**Assumption 1** (Regularity of Distribution (Wan et al., 2024)). We assume the data distribution $p$ satisfies the following regularity conditions:

- It has finite 2-moment, i.e., $\int_{\mathbb{R}^n} \|x\|^2 p(\mathrm{d}x) < \infty$;

- Its support, $\Omega = \mathrm{supp}(p)$, has a positive reach;

- It has a non-vanishing density, i.e., there exist constants $k \geq 0$ and $c > 0$ such that for any radius $R > 0$, there exists a constant $C_R > 0$ where for any small radius $0 \leq r < c$ and any $x \in B_R(0) \cap \Omega$, we have $p(B_r(x)) \geq C_R r^k$.

The finite 2-moment requirement is standard in the literature for analyzing generative models. The positive reach condition is satisfied in our setting since we consider a convex support set $\Omega = \mathcal{C}$. The last requirement ensures the density does not vanish within the support, which prevents pathological cases where probability mass concentrates on lower-dimensional manifolds.

There are also other (more restrictive) regularity conditions for the data distribution in the literature, including the covariance condition (Benton et al., 2023) and $\kappa$-semi-log-concave/convex (Gao et al., 2024a), and convex support set (Gao et al., 2024b). It remains largely open for the minimal necessary conditions to establish the well-posedness of flow matching models.

**Assumption 2** (Neural Network Approximation Properties). Given a distribution $p$ with support $\Omega$, let $u(x, t)$ be the designed target vector field and $v_\theta$ be the neural network approximated vector field in the flow-matching model.

- $v_\theta$ is $L_\theta$-Lipschitz for $x \in \Omega$ and $t \in [0, 1]$;

- The $\ell_2$ approximation error is bounded as $\epsilon_\theta^2 = \mathbb{E}_{x_t, t} \|v_\theta(x_t, t) - u(x_t, t)\|^2$, where $p_t$ is the probability density driven by the target vector field $u$.

The Lipschitz constant of a trained neural network is bounded within a compact set in our setting $\Omega = \mathcal{C}$. The training loss in Eq. (1) is equivalent to the vector field approximation error up to a constant (Lipman et al., 2022). Therefore, we can minimize the approximation error through proper training of the loss function.

**Lemma 2** (Error Bound for Flow Matching (Benton et al., 2023)). *For the vanilla flow matching model:* $x_1 = x_0 + \int_0^1 v_\theta(x, t)\, \mathrm{d}t$, *with induced probability distribution $p_\theta$ at $t = 1$. The Wasserstein-2 distance between the data distribution $p_{data}(x)$ and the approximated distribution $p_\theta(x)$ is bounded by*

$$\mathcal{W}_2(p_{data}(x), p_\theta(x)) \leq \mathrm{e}^{L_\theta} \epsilon_\theta \tag{43}$$

**Lemma 3** (Error Bound for Reflected Flow Matching (Xie et al., 2024)). *For the reflected flow matching model over the convex domain $\mathcal{C}$:* $x_1 = x_0 + \int_0^1 v_\theta(x, t) + R(x_t)\, \mathrm{d}t$, *with induced probability distribution $p_\theta^r$ at $t = 1$. The Wassserstein-2 distance between the data distribution $p_{data}(x)$ and the approximated distribution $p_\theta^r(x)$ is bounded by*

$$\mathcal{W}_2(p_{data}(x), p_\theta^r(x)) \leq \mathrm{e}^{1/2 + L_\theta} \epsilon_\theta \tag{44}$$

### D.2    Proof of Regularity of GFM in Prop. 5.1

*Proof.* Given the original data distribution $p$ over compact convex set $\Omega_p = \mathcal{C}$ satisfying the regularity conditions; We verify the three regularity conditions one by one for the transformed data distribution $q = \Phi_\#^{-1} p$ over a unit ball $\Omega_q = \mathcal{B}$, where $\Phi$ is a bi-Lipschitz homeomorphism (e.g., gauge mapping).

- **Finite second moment:** since $q$ has bounded support, it immediately implies a finite second moment.

- **Positive reach:** The support of $q$ is a unit ball $\Omega_q = \mathcal{B}$, which has infinite positive reach. We remark that even if the original support $\Omega_p$ does not have a positive reach (e.g., star-shaped set), after transformation, the $\Omega_q$ is always a unit ball and satisfies this regularity condition on the reach.

- **Non-vanishing density:** Given the bi-Lipschitz property, we have:

$$L_1 \|z_1 - z_2\| \leq \|\Phi(z_1) - \Phi(z_2)\| \leq L_2 \|z_1 - z_2\|, \quad \forall z_1, z_2 \in \mathcal{B} \tag{45}$$

where $L_1 = 1/L_{\Phi^{-1}}$ and $L_2 = L_\Phi$ are the (inverse) Lipschitz constants.

For any $z \in \Omega_q \cap B_R(0)$ and any small radius $r < \frac{c}{L_2}$, where $c$ is from the non-vanishing density property of $p$, we have:

$$q(B_r(z)) = p(\Phi(B_r(z))) \tag{46}$$
$$\geq p(B_{L_1 r}(\Phi(z))) \tag{47}$$

since $\Phi(B_r(z)) \supseteq B_{L_1 r}(\Phi(z))$ by the bi-Lipschitz property.

Since $\Phi$ is bi-Lipschitz, for any $R > 0$, there exists $R' > 0$ such that $\Phi(B_R(0) \cap \Omega_q) \subseteq B_{R'}(0) \cap \Omega_p$. By the non-vanishing density property of $p$, there exists a constant $C_{R'} > 0$ such that $p(B_{L_1 r}(\Phi(z))) \geq C_{R'}(L_1 r)^k$.

Therefore, we have

$$q(B_r(z)) \geq C_{R'} L_1^k r^k \tag{48}$$

This establishes the non-vanishing density property for $q$.

$\square$

## D.3 Proof of Error Bound of GFM in Prop. 5.2

*Proof.*

$$\mathcal{W}_2^2(p_{data}, p_\theta^{gr}) = \inf_{\gamma = \Pi(p_{data}, p_\theta^{gr})} \left\{ \int \|x_1 - x_2\|^2) d\gamma \right\} \tag{49}$$

$$= \inf_{\gamma = \Pi(q_{data}, q_\theta^{gr})} \left\{ \int \|\Phi(z_1) - \Phi(z_2)\|^2) d\gamma \right\} \tag{50}$$

$$\leq L_\Phi^2 \inf_{\gamma = \Pi(q_{data}, q_\theta^{gr})} \left\{ \int \|z_1 - z_2\|^2) d\gamma \right\} \tag{51}$$

$$\leq L_\Phi^2 \mathcal{W}_2^2(q_{data}, q_\theta^r) \tag{52}$$

$$\leq L_\Phi^2 e^{1+2L_\theta} \epsilon_\theta^2 \tag{53}$$

$\square$

where Eq. (50) is by the equivalence of the distributions under a homeomorphic push-forward mapping: $p_{\text{data}} = \Phi_{\#} q_{\text{data}}$ and $p_{\text{data}}^{gr} = \Phi_{\#} q_{\text{data}}^{qr}$; Eq. (51) is by the Lipschitz property of gauge mapping $\Phi$ shown in Prop. 4.1; Eq. (53) is the error bound of the regular reflected generation over the unit ball under Lemma 3.

**Remark**. The Lipschitz constant of gauge mappings between compact sets remains bounded inherently. This property stands in contrast to mirror mapping-based generative models (Liu et al., 2024b), which map open convex sets to $\mathbb{R}^n$. In the latter case, the Lipschitz constant can grow unbounded as points near the boundary are mapped to infinity, significantly complicating approximation error analysis. Our Gauge Flow Matching circumvents this limitation, providing theoretical guarantees on the Wasserstein-2 distance between the learned and data distributions. To optimize the model's performance, we can further reduce the Lipschitz constant of the gauge mapping by identifying an interior point $x^\circ$ that serves as the "center" of the constraint set.

## D.4 Proof of Generation Complexity of GFM in Prop. 5.3

*Proof.* Closed-form computation: Based on Table 8, the algorithmic complexity of naturally derived under different constraint functions.

Bisection-based computation: Based on algorithm 3, the algorithmic complexity for the bisection algorithm needs the feasibility check $x \in \mathcal{C}$, which essentially involves calculating the constraint function $g_i(x)$ for $i = 1, \ldots, m$. The number of bisection steps needed to achieve a target error is derived as $\mathcal{O}(\text{diam}(\mathcal{C}) \cdot \log(1/\epsilon_{\text{bis}}))$.

$\square$

## E Experimental Settings

In this section, we describe the experimental setup used to generate the results reported in Section 7. Our proposed model is implemented in PyTorch (Paszke et al., 2019), with all models trained using Adam optimizer (Kingma & Ba, 2014) with hyperparameters $\beta_1 = 0.99$, $\beta_2 = 0.999$, and a learning rate of $10^{-3}$. For sample generation, we solve the forward ODE using the method proposed by (Chen

et al., 2018), and we follow the approach in (Xie et al., 2024) to solve the reflected ODE in Equation (5).

### E.1 CONSTRAINED GENERATIVE MODELS BASELINES

We compare our approach against the following state-of-the-art constrained generative models, selected for their ability to handle various constraint types:

- **FM**: Vanilla flow matching that transforms a Gaussian distribution to the target distribution using linear conditional flow (Lipman et al., 2022; Liu et al., 2022b).
- **DM**: Vanilla diffusion model with variance-preserving diffusion process (Ho et al., 2020; Song et al., 2020).
- **Reflection**: A method that applies reflection terms when generated samples encounter constraint boundaries (Xie et al., 2024).
- **Metropolis**: Metropolis Sampling approach for approximating reflection-based generation (Fishman et al., 2024).
- **Projection**: An approach utilizing orthogonal projection when generated samples violate constraints (Christopher et al., 2024).
- **GFM**: Our proposed framework as detailed in Section 4.

We remark that all models share the same training settings (dataset, optimizers, hyperparameters) and generation settings (ODE methods, step size). For Reflection, Metropolis, and Projection models, the same velocity models are used. Therefore, they share the same training time as reported in the Table. 5.

The additional constraint-handling mechanisms (e.g., reflection and projection) are implemented based on references. We emphasize that closed-form projection operators generally do not exist for the complex constraints in our experiments, such as intersections of polytopes and ellipsoids, and semidefinite cone constraints. Therefore, we employ state-of-the-art optimization solvers: (i) MOSEK for convex projection and (ii) IPOPT for non-convex cases. While these are highly optimized solvers, iteratively solving a constrained optimization problem at each sampling step for each sample (whenever the ODE trajectory violates constraints) remains computationally expensive. This cost is particularly significant for high-dimensional problems or tight constraint sets where violations occur frequently throughout the sampling process.

We also remark that, in the geodesic-convex set generation tasks, the FM and DDPM models are trained to generate samples in the tangent space of a prescribed point, which is the same across all methods.

### E.2 REFLECTION COMPUTATION

We follow the notations in (Xie et al., 2024) and (Lou & Ermon, 2023). Let $\boldsymbol{L}_t$ be the reflection term that reflect the outward velocity at the boundary $\partial \mathcal{C}$. Given an initial point $x_{\text{init}}$, the reflected ordinary differential equation is

$$\mathrm{d}x_t = v(x_t, t)\mathrm{d}t + \mathrm{d}\boldsymbol{L}_t, \tag{54}$$

$$x_0 = x_{\text{init}}. \tag{55}$$

Intuitively, the reflection term $\boldsymbol{L}_t$ in equation (54) pushes the trajectory back to the domain $\mathcal{C}$ once the trajectory hits the boundary. Under mild conditions, the solution to the reflected ODE exists and is unique (Xie et al., 2024).

Empirically, the reflected ODE can be solved by numerical solvers, for example, using the Euler method. The Euler method iteratively compute the trajectory points by

$$x_{t+\delta t} = x_t + \delta t \cdot v(x_t, t), \tag{56}$$

where $t \in [0, 1)$ is an intermediate time and $\delta t > 0$ is a small time step. Euler step with reflection can be computed as

$$x_{t+\delta t} = x_t + \delta t \cdot \left[ v(x_t, t) - 2(1 - s_t)\big(v(x_t, t)\big)^T \boldsymbol{n}_{\partial \mathcal{C}} \boldsymbol{n}_{\partial \mathcal{C}} \right] \tag{57}$$

---

**Algorithm 4** Euler method with reflection

---

**Input:** A velocity field $v(x_t, t)$, a domain $\mathcal{C}$, an initial point $x \in \mathcal{C}$, number of steps $N$, a function that gives outward normal vector $\boldsymbol{n}_{\partial \mathcal{C}}(x)$, and a function that computes the distance to boundary $s_{\partial \mathcal{C}}(x_t, v_t)$.

1: Let $\delta t = 1/N$, $x_0 = x$
2: **for** $i = 0, 1, 2, \ldots, N-1$ **do**
3:     let $t = i/N$, $v_i = v(x_i, t)$
4:     update $x_{i+1} = x_i + \delta t \cdot v_i$
5:     **if** $x_{i+1} \notin \mathcal{C}$ **then**
6:         compute distance to boundary $s_i = \min\{1, s_{\partial \mathcal{C}}(x_i, \delta t \cdot v_i)\}$
7:         compute boundary point $x_i' = x_i + s\delta t \cdot v_i$
8:         compute normal vector $n_i = \boldsymbol{n}_{\partial \mathcal{C}}(x_i')$
9:         compute reflection term $L_i = -2\delta t(1 - s_i)v_i^T n_i n_i$
10:      update $x_{i+1} = x_{i+1} + L_i$
11:     **end if**
12: **end for**
**Output:** $x_N$

---

Table 9: Closed-form Expressions for Distance and Normal Vector Functions

| Constraints | Formulation | Distance Function | Normal Vector |
|---|---|---|---|
| Unit Ball | $\|x\|_2 \le 1$ | $s(x, v) = \{\text{root}(v^T v, 2v^T x, x^T x - 1)\}^+$ | $\boldsymbol{n}(x) = x/\|x\|$ |
| Unit Cube | $\|x\|_\infty \le 1$ | $s(x, v) = \min_{i=1,2,\ldots,n}\{(1 - \text{sign}(v_i)x_1)/v_i\}$ | $\boldsymbol{n}(x) = \mathbf{e}_{\arg\max_{i=1,2,\ldots,n} |x_i|}$ |
| Linear | $a^T x \le b$ | $s(x, v) = (b - a^T x)/a^T v$ | $\boldsymbol{n}(x) = a/\|a\|$ |
| Quadratic | $x^T Q x + a^T x \le b$ | $s(x, v) = \{\text{root}(A_Q, B_Q, C_Q)\}^+$ | $\boldsymbol{n}(x) = -2Qx - a$ |

[1] Notation: $x, a \in \mathbb{R}^n, b \in \mathbb{R}, Q \in \mathbb{S}_+^n, A \in \mathbb{R}^{n \times k}, p \in \mathbb{R}^k, F_0, \cdots, F_n \in \mathbb{R}^{k \times k}$.

[2] $\boldsymbol{e}_1 = (1, 0, 0, \ldots, 0), \boldsymbol{e}_2 = (0, 1, 0, 0, \ldots, 0), \ldots, \boldsymbol{e}_{n-1} = (0, 0, \ldots, 0, 1, 0), \boldsymbol{e}_n = (0, 0, \ldots, 0, 1)$.

[3] $A_Q = v^\top Q v$, $B_Q = 2x^{\circ\top} Q v + a^\top v$, $C_Q = x^{\circ\top} Q x^\circ + a^\top x^\circ - b$.

[4] $(\cdot)^+ = \max(\cdot, 0)$.

[5] $\text{root}(x_1, x_2, x_3) = \frac{-x_2 \pm \sqrt{x_2^2 - 4x_1 x_3}}{2x_1}$ denotes the quadratic equation solution

where $s_t = \min\left\{1, \inf\{s > 0 \mid x_t + s\delta t \cdot v(x_t, t) \notin \mathcal{C}\}\right\}$ is the distance from $x_t$ to the nearest boundary $\partial \mathcal{C}$ along direction $v(x_t, t)$, and $\boldsymbol{n}_{\partial \mathcal{C}}$ is the outward normal vector at $x_t + s\delta t \cdot v(x_t, t) \in \partial \mathcal{C}$. Algorithm 4 summarizes the Euler method with reflection and table 9 lists reflections for several common convex sets.

### E.3 PERFORMANCE EVALUATION METRICS

**Feasibility ratio computation** Suppose that $x_1, x_2, \ldots, x_N$ are generated samples, $g_i(x), i = 1, 2, \ldots, m$ are constraints of interest. One sample $x$ is feasible if

$$g_i(x) \le 0, i = 1, 2, \ldots, m. \tag{58}$$

The feasibility ratio of the batch samples is defined as

$$\text{feasibility}\{x_1, x_2, \ldots, x_N\} = \frac{\#\{x_i \mid i = 1, 2, \ldots, N, g_k(x_i) \le 0, k = 1, 2, \ldots, m\}}{N}. \tag{59}$$

In our experiments, we set the tolerance as $10^{-6}$.

### E.4 CONSTRAINT FORMULATIONS IN ILLUSTRATIVE EXAMPLES

We first give the specific formulations of those constraints:

- **Convex Set**:

$$\{x \in \mathbb{R}^2 \mid Ax \le b, \|x - c\|_2 \le 2.5, x^T Q x + p^T x + d \le 0\} \tag{60}$$

where $A, b, c, Q, p, d$ are randomly sampled.

- **Star-convex Set**:

$$\{x \in \mathbb{R}^2 \mid \|x\| \leq \Gamma_{\alpha,n}(x) := 1 + \alpha \sin(n \arctan(x_2/x_1))\} \tag{61}$$

where $\alpha \in (0, 1)$ and $n \in \mathbb{Z}_+$ determine the size and number of stars, respectively.

- **Geodesic-convex Set**:

$$\{x \in \mathbb{S}^2 \mid Ax \leq b\} \tag{62}$$

where $\mathbb{S}^2$ is the sphere in 3-dimensional Euclidean space and is constrained by a set of linear inequalities.

The data are sampled from a mixed Gaussian distribution, whose location parameter and covariance are generated randomly.

**Model settings:** We model the velocity field with 2 hidden layers with exponential linear unit (ELU) activation functions. We train the models for 10000 epochs with a batch size of 256, and prior distribution as the uniform distribution over the constrained/ball domain. Samples are generated by the Euler algorithm in 1,000 steps. For the reflected/projected methods, additional reflection/projection is performed after each step.

### E.5 MIRROR MAP *vs* GAUGE MAP ON SIMPLEX DOMAIN

We compare our GFM model against several baseline methods: Vanilla Flow Matching, DDPM, and Mirror Diffusion. We consider the standard simplex in $d$ dimensions:

$$S_d = \{x \in \mathbb{R}^d \mid x_i \geq 0, \sum_{i=1}^{d} x_i \leq 1\}. \tag{63}$$

**Data preparation:** We generate training data from a mixture of multivariate normal distributions, with each component centered near a distinct geometric feature of the simplex. The mixture includes $d$ vertex-centered modes, one origin-centered mode, and one face-centered mode. The location parameters are defined as:

$$\mu_i = (\underbrace{0.1/d, \ldots, 0.1/d}_{i}, 0.9, \underbrace{0.1/d, \ldots, 0.1/d}_{d-i-1}), \quad i = 1, 2, \ldots, d,$$
$$\mu_o = (0.1/d, \ldots, 0.1/d),$$
$$\mu_f = (0.9/d, \ldots, 0.9/d).$$

Here, $\mu_i$ places most of its mass at the $i$-th vertex, $\mu_o$ is centered near the origin, and $\mu_f$ distributes mass in the simplex face. We generate a total of 10,000 training samples from this mixture.

**Model settings:** For illustrative 2-dimensional cases, we modeled the time-variant velocity field using 3 layers with 256 units each and ELU activation functions. For 50-dimensional cases, we modeled the time-variant velocity field with 4 hidden layers of 512 hidden units, incorporating residual connections and a bottleneck structure (Liu et al., 2024a).

As shown in Table 10, in the synthetic simplex domain example, our GFM method achieves comparable performance to the vanilla flow matching baseline and other constrained generation baselines in low-dimensional settings ($d = 2$), while achieving significantly faster inference speed. In high-dimensional settings ($d = 50$), GFM attains the best fidelity to the data distribution while maintaining faster inference speed. It is worth noting that the Mirror Map model (Liu et al., 2024b) exhibits fast performance in both training and inference phases. This is because a low-complexity, closed-form mirror map exists for this simple simplex domain. However, the mirror map approach fails for more complex constraints, such as the PSD cone constraint.

### E.6 CONFIGURATION GENERATION IN ROBOTIC CONTROL BENCHMARK

We follow the procedure reported by (Jaquier et al., 2021) to learn the trajectories of the manipulability ellipse. In planar letter drawing problems, the manipulability ellipses are modeled by SPD matrices

Table 10: Results for synthetic generation tasks over the simplex domain.

|  | Method | FM | DDPM | Mirror Map | **GFM** |
|---|---|---|---|---|---|
| $d = 2$ | Feasibility (%) | 94.5 | 93.8 | 100 | **100** |
|  | MMD ($\times 10^{-3}$) | 6.26 | 14.02 | 8.34 | 6.53 |
|  | Training (s) | 0.09 | 0.09 | 0.13 | 0.13 |
|  | Inference (s) | 0.40 | 0.94 | 0.46 | 0.97 |
| $d = 50$ | Feasibility (%) | 11.7 | 4.35 | 100 | **100** |
|  | MMD ($\times 10^{-2}$) | 9.08 | 1.82 | 6.00 | 2.56 |
|  | Training (s) | 0.14 | 0.13 | 0.14 | 0.15 |
|  | Inference (s) | 0.93 | 1.20 | 0.93 | 1.38 |

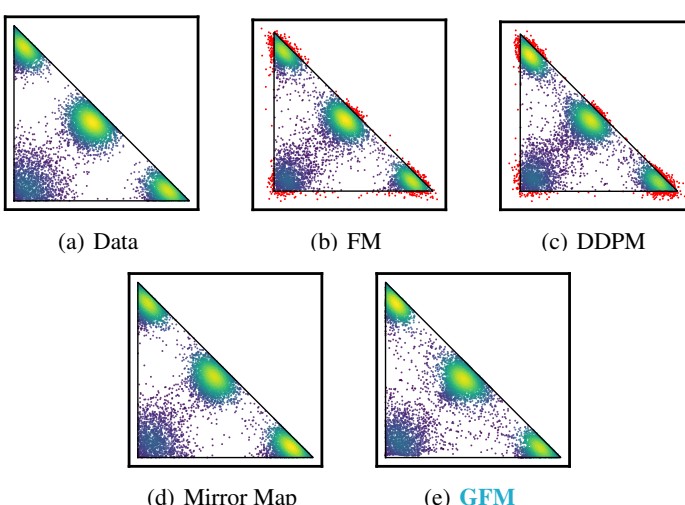

(a) Data         (b) FM         (c) DDPM

(d) Mirror Map         (e) **GFM**

$M = \{X \in \mathbb{S}^{2 \times 2} \mid X \succeq 0, \ \mathrm{tr}(X) \leq C\}$. In our formulation, we transform it into the equivalent linear matrix inequality formulation:

$$M = \left\{ x \in \mathbb{R}^3 \mid x_1 \begin{pmatrix} 1 & 0 \\ 0 & 0 \end{pmatrix} + x_2 \begin{pmatrix} 0 & 1 \\ 1 & 0 \end{pmatrix} + x_3 \begin{pmatrix} 0 & 0 \\ 0 & 1 \end{pmatrix} \succeq 0, \quad \mathrm{tr}(M) = x_1 + x_3 \leq C \right\} \quad (64)$$

We additionally learn to sample the 2-dimensional trajectories. Therefore, our models are parameterized to generate samples in $\mathbb{R}^2 \times M$

**Model settings:** We model the time-variant velocity field with 3 hidden layers with 256 units each and ELU activation functions. We train the models for 10,000 epochs with a batch size of 256, and prior distribution as the uniform distribution over the constrained/ball domain.

**Remark:** The feasibility ratios reported in Table 11 are also high for methods that do not force feasibility, e.g., vanilla flow-matching and diffusion models. The possible reason for that phenomenon is that the training data is highly concentrated in the interior of the domain, and only a few data points are close to the boundary. Therefore, a few generated samples are infeasible.

### E.7 SOLUTION GENERATION FOR RELAXED COMBINATORIAL PROBLEMS

We further evaluate our approach on high-dimensional solution generation for relaxed combinatorial problems (Kook & Vempala, 2024). The distribution we consider is particularly interesting as it recovers, as special cases, the sampling problems associated with Max-Cut SDP relaxations and minimum volume ellipsoid problems, defined as:

$$X \sim \exp\left(-(\langle A, X \rangle + \|X - B\|_F^2 + \|X - C\|_F - \log \det X)\right) \quad (65)$$

$$\text{s.t. } X \succeq 0, \langle D_i, X \rangle \geq c_i, \quad \forall i \in [m]. \quad (66)$$

Table 11: Results for robotic manipulability ellipse generation task.

| Methods | FM | DDPM | **GFM** |
|---|---|---|---|
| Feasibility (%) | 99.92 | 99.94 | **100** |
| MMD ($\times 10^{-2}$) | 5.39 | 5.58 | 9.85 |
| Training (s) | 3.43 | 3.64 | 3.55 |
| Inference (s) | 0.50 | 0.86 | 0.83 |

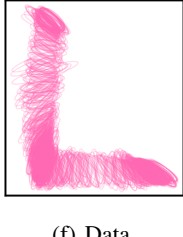 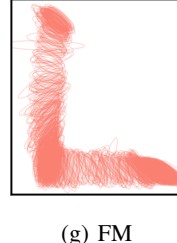 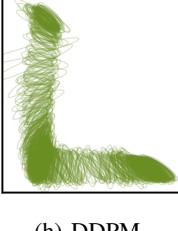 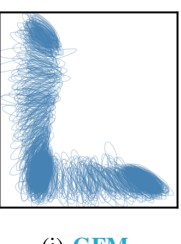

(f) Data      (g) FM      (h) DDPM      (i) GFM

Figure 11: Visual comparison of 1000 samples between data distribution, DDPM model, Flow Matching model, and our GFM model on robotic manipulability ellipse generation task.

The constraints ensure that $X$ is a positive semidefinite $n \times n$ matrix satisfying $m$ linear inequalities. Each term in the energy function captures different structural aspects of the optimization landscape, including linear objectives, quadratic penalties, proximity constraints, and determinant-based regularization.

By developing generative models for this class of problems, we aim to provide tools that not only sample feasible solutions but also help explore the solution space in ways that could reveal insights about the underlying combinatorial structure. This capability is particularly valuable for understanding the landscape of near-optimal solutions and for generating diverse candidate solutions that might be refined by downstream processes.

**Data preparation:** We sample from the target distribution using the Hit-and-Run (Bélisle et al., 1993) algorithm. The Hit-and-Run sampling algorithm is a Markov Chain Monte Carlo (MCMC) method that generates random samples from a high-dimensional distribution by iteratively selecting a random direction and moving to a new point along that direction according to the target distribution. To generate training samples, we use the Hit-and-Run sampler with 1000 burn-in, and sample new points using 1-dimensional Metropolis sampling with 100 burn-in.

**Model settings:** We model the time-variant velocity field with 3 hidden layers with $n^2/2$ hidden units with residual connections and bottleneck structure (Liu et al., 2024a). We train the models for 1000 epochs with a batch size of 256, and prior distribution as the uniform distribution over the constrained/ball domain.

### E.8 CONSTRAINED CONDITIONAL TIME SERIES GENERATION

We follow the experimental setup of (Narasimhan et al., 2025) to generate time-series data subject to physical and statistical constraints:

$$\underbrace{x_{\min} \leq x_t \leq x_{\max}}_{\text{capacity bounds}}, \quad \underbrace{\mu_{\min} \leq \text{ave}(x_{1:T}) \leq \mu_{\max}}_{\text{mean volume bounds}}, \quad \underbrace{\text{var}(x_{1:T}) \leq \sigma_{\max}^2}_{\text{variance bounds}}. \quad (67)$$

These constraint bounds are derived from empirical statistics of the historical dataset. The time series is generated via conditional forecasting, using past observations as model input.

**Data preparation:** We use the PEMS-BAY traffic dataset (Yoon et al., 2019; Li et al., 2018), partitioned by days. The models learn to generate time series for one day conditioned on data from the previous day. Days with incomplete data are excluded from the dataset.

**Model settings**: We model the conditional time-variant velocity field using a 4-layer neural network with 64 hidden units per layer, incorporating residual connections and a bottleneck structure (Liu et al., 2024a). All models are trained for 4,000 epochs with a batch size of 180. For the prior distribution, we use a standard Gaussian for vanilla Flow Matching and projection-based methods, and a uniform distribution over the ball domain for GFM.

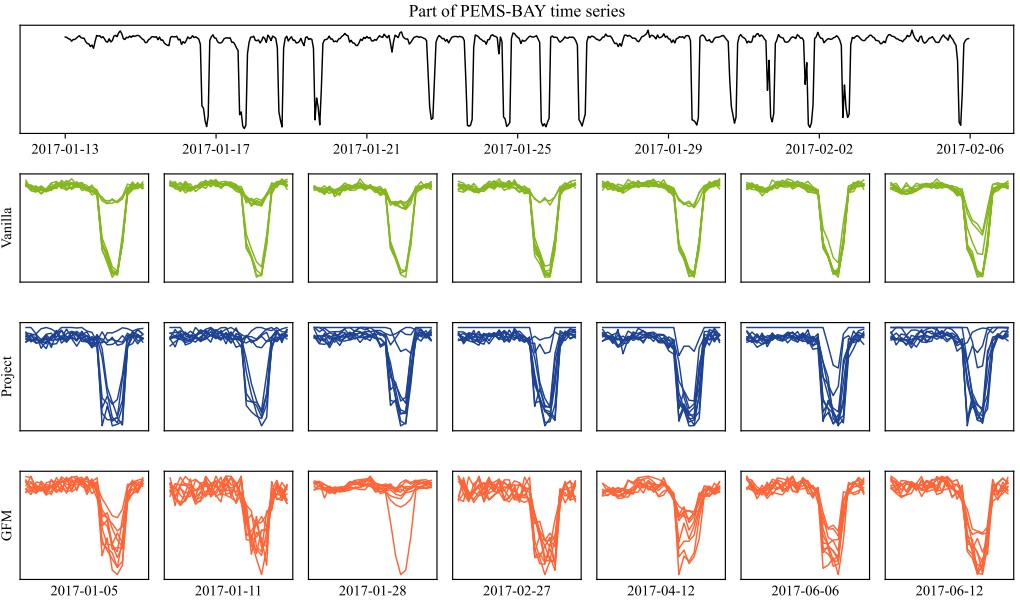

Figure 12: Visual comparison of 24 time series strides between data distribution, Vanilla Flow Matching model, Projection Flow Matching model, and our GFM model on constrained traffic time series generation task.

### E.9 WATERMARKED IMAGE GENERATION TASKS

We evaluate GFM on constrained image generation with embedded watermarks using CIFAR-10, which consists of $32 \times 32$ RGB images. Following the watermarking methods in (Liu et al., 2024b), we partition pixels $\mathbf{x} \in [0,1]^d$ into unconstrained public pixels $\mathbf{x}_1$ and watermark pixels $\mathbf{x}_2 \in \mathcal{C}$, where $\mathcal{C}$ is a randomly sampled polytope for watermarking.

**Model settings:** All baselines are distilled from a publicly available checkpoint of flow matching models (Tong et al., 2024), which share the same U-Net architecture (34M parameters). All models use identical distilled hyperparameters (100 epochs) and sampling settings (100-step Euler method) on CIFAR-10 for fair comparison. We follow the method in (Salimans & Ho, 2022) to train the distilled models by minimizing the MSE from the teacher model:

$$\mathcal{L}_{\text{distill}}(\phi) = \mathbb{E}_{t \sim U(0,1), x_0 \sim p_0} \| v_\phi(x_t(x_0), t) - v_\theta(x_t(x_0), t) \|_2^2, \tag{68}$$

where $x_t$ is obtained by integrating teacher ODE from $x_0$ with step length $0.001$.

**Data preparation:** We select 32 pixels as watermarks constrained via a random polytope $\mathcal{C} = \{\mathbf{x}_2 : A\mathbf{x}_2 \leq b\}$ with $b > 0$. The watermarked training dataset is created by projecting the selected pixels in the original CIFAR-10 dataset onto the polytope $\mathcal{C}$, while other public pixels remain unchanged.

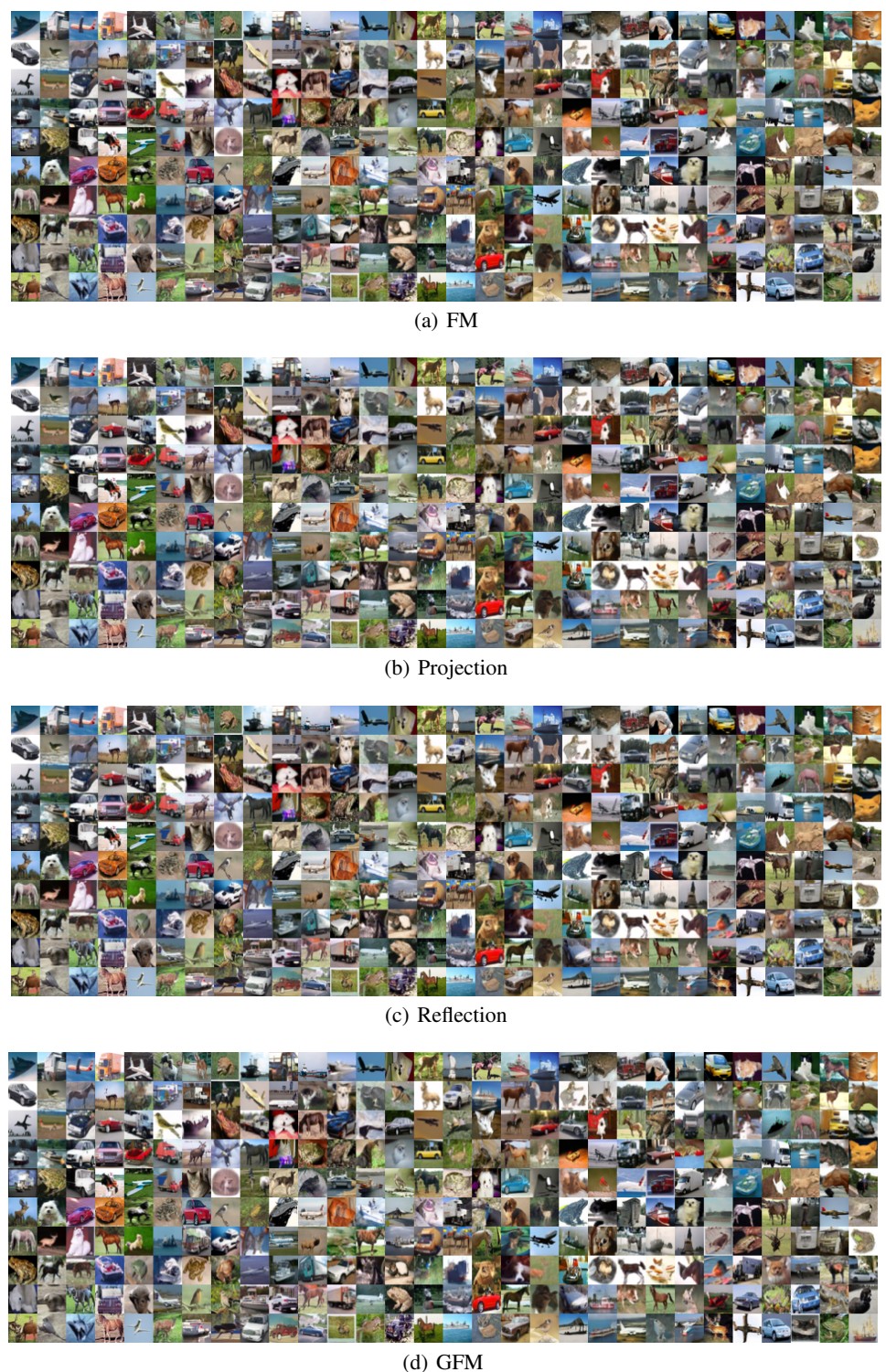

(a) FM

(b) Projection

(c) Reflection

(d) GFM

Figure 13: CIFAR-10 samples with embedded watermarks generated by various methods using 100-step Euler sampling.

E.10 Setting of Scalability Tests of Gauge Mapping

To evaluate the scalability of gauge mapping, we conducted experiments with randomly generated inequality constraints. We tested scalability across three dimensions: problem dimension, batch size, and number of constraints.

We established a baseline configuration with 500-dimensional variables ($n = 500$), constraint dimension $k = 500$, $m = 8$ constraints, and batch size 256. For each scalability test, we varied one parameter while keeping the others fixed at their baseline values:

- **Problem dimension scalability:** We varied $n \in [10, 3000]$ while fixing $k = 500$, $m = 8$, and batch size 256. The specific dimensions tested follow Table 8.

- **Batch size scalability:** We varied the batch size from 10 to 100,000 while fixing $n = 500$, $k = 500$, and $m = 8$.

- **Constraint scalability:** We varied the number of constraints $m \in [8, 512]$ while fixing $n = 500$, $k = 500$, and batch size 256.

For each configuration, we measured the computation time of the gauge function evaluation. The results are illustrated in Fig. 8.

To demonstrate the efficiency of the bisection algorithm (Algorithm 3), we tested gauge computation on convex high-dimensional polynomials generated using the sum-of-squares (SOS) method.

A sum-of-squares polynomial of $n$ variables and degree $d$ is defined as

$$p(x_1, x_2, \ldots, x_n) = \big(\mathrm{mono}_d(x_1, x_2, \ldots, x_n)\big)^T Q, \mathrm{mono}_d(x_1, x_2, \ldots, x_n), \tag{69}$$

where $\mathrm{mono}_d(x_1, x_2, \ldots, x_n)$ is a vector of all monomials up to degree $d$, and $Q$ is a positive semi-definite matrix. Each monomial of degree $k$ takes the form $x_1^{c_1} x_2^{c_2} \cdots x_n^{c_n}$ where $c_1, c_2, \ldots, c_n \in \mathbb{N} \cup 0$ and $c_1 + c_2 + \cdots + c_n = k$.

We performed the bisection-based gauge function, and the results are summarized in Table 12.

Table 12: Gauge function computation time of high-dimensional polynomials using bisection

| No. of polynomials | No. of variables | Degree | No. of monomials | Time (s) |
|---|---|---|---|---|
| 10 | 10 | 4 | $4,356$ | 0.025 |
| 10 | 50 | 4 | $1,758,276$ | 0.492 |
| 10 | 100 | 4 | $26,532,801$ | 5.592 |

E.11 Ablation Study

We consider the following ablation study to examine key components of GFM, including

- **Impacts of the interior point selection**: Central interior point *vs* Near-boundary interior point

- **Impacts of different generation strategies**: Reflection-based sampling *vs* Projection-based sampling

- **Impacts of the initial set**: $\ell_2$-norm ball *vs* $\ell_\infty$-norm cube

We follow a similar setting in Sec. 7.1.

Table 13: Performances on joint convex set under different settings.

| | | $L_2$ | | $L_\infty$ | |
| --- | --- | --- | --- | --- | --- |
| | | Central | Boundary | Central | Boundary |
| Reflection | MMD ($\times 10^{-2}$) | 0.34 | 43.50 | 0.35 | 43.59 |
| | Training (s) | 0.22 | 0.22 | 0.21 | 0.23 |
| | Inference (s) | 0.91 | 0.86 | 0.86 | 0.88 |
| Projection | MMD ($\times 10^{-2}$) | 3.14 | 37.82 | 3.34 | 38.13 |
| | Training (s) | 0.22 | 0.22 | 0.21 | 0.23 |
| | Inference (s) | 0.48 | 0.50 | 0.47 | 0.50 |

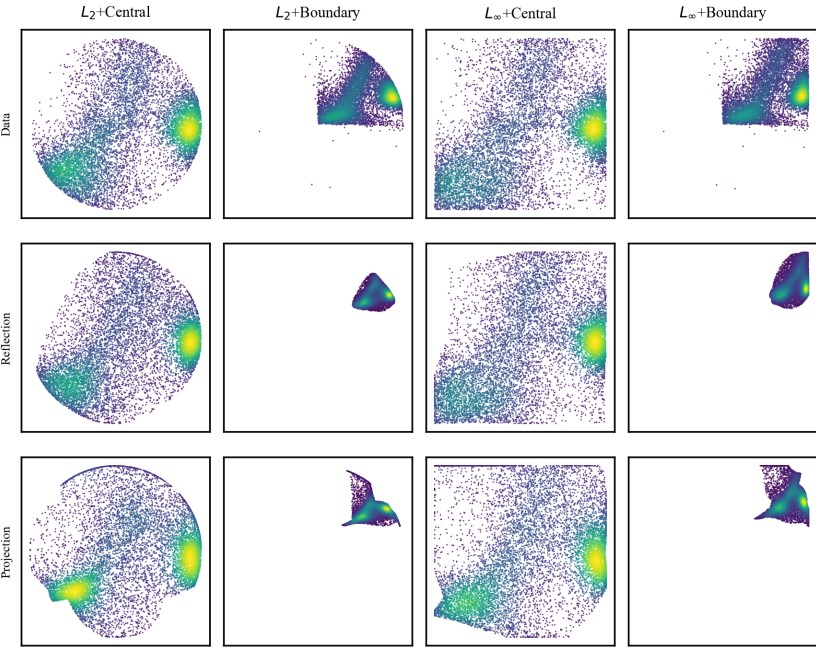

Figure 14: Visual comparison in the transformed space over $L_2$-norm **ball** and $L_\infty$-norm **cube**: We compare gauge mappings using **central** versus **near-boundary** interior points, and evaluate samples generated via projection versus reflection-based strategies. Results show that gauge mapping with central interior points reduces distortion in the transformed space, facilitating more effective GFM training. Reflection-based sampling achieves superior generation quality compared to projection methods, while the choice of initial set shows negligible impact on generation quality.

