# OpenReview forum: "Gauge Flow Matching: Efficient Constrained Generative Modeling over General Convex Set and Beyond"
_ICLR.cc/2026/Conference — ICLR 2026 Poster_

### Official Review · Reviewer_rKAZ · 2025-10-15

**Soundness:** 4
**Presentation:** 4
**Contribution:** 4
**Rating:** 8
**Confidence:** 3

**Summary:**

This paper addresses a critically important and timely challenge of constrained generative modeling by proposing a novel idea called gauge flow matching (GFM). The underlying idea of GFM is to introduce a bijective gauge mapping to transform a complex constrained generative modeling problem into an equivalent modeling problem over a simple unit ball domain. The paper presents rigorous theoretical validation/analysis on the strict constraint guarantee and a lower computational complexity on not only convex feasibility regions but also certain types of non-convex feasibility regions. It also presents empirical validation of the GFM idea on various benchmarks, in comparison to other state-of-the-art constrained generation methods. The proposed idea is technically sound, and the scientific breakthrough is clear. The presentation and the style of writing are also effective, maintaining a good balance between high-level intuitive explanation and low-level mathematical details. As such, I believe this paper deserves presentation at ICLR.

**Strengths:**

Although the idea of bijective mappings to transform constrained domains into a new domain that has more favorable analytical properties existed before this paper, it was still understudied, and the previous work was largely limited to some limited problems and/or theoretical guarantees. Hence, I find the idea in this paper sufficiently original and of significance, presenting creative combinations of existing ideas (e.g., guage mapping + bijective mapping to simplify constrained generation), new scientific knowledge (e.g., theoretical guarantees, complexity analysis, benchmark test results), and such.

I also appreciate the style and quality of writing. While engaging a broader group of audience effectively with a clear and intuitive explanation of the method and the rationale behind the method, the paper still presents enough technical details with mathematical rigor.

**Weaknesses:**

First and foremost, I would've appreciated more experiments on challenging generative tasks. I understand that the authors did their best to cover the spectrum of discussions from theoretical to empirical validations, the real-world generative tasks (i.e., robotic manipulability and relaxed combinatorial problems), but I still feel underwhelmed when it comes to the empirical validation portion of this work. Image generation, for example, would've been a great benchmark test, even if the scope of images was limited to a small, limited variety (e.g., microscopic images as opposed to the whole set of natural images). I understand such a task might be "too non-convex" and might be beyond the scope of this work, but I still remain curious to know if and how far the proposed GFM idea could push the limit. (Especially because the other parts of the works are very well done!)

Other than that, I only have some minor suggestions regarding mostly presentation, which the authors may or may not want to consider:
- I think Figure 1 captures the core idea of your work very well. Why don't you move it to an earlier page?
- Also, figure captions in general could be more elaborate and contain more details.
- Excessive use of bold faces and color highlights hurts readability in my opinion. As an example (out of other such cases), I don't see the need to emphasize the acronym "GFM" and the word "limitation" in the conclusion section by making them boldface. This tendency is found across the whole paper, which the authors are strongly encouraged to revisit and think through.

**Questions:**

I don't have questions.

---

> ### Author Response · Authors · 2025-11-27
> **Response**
>
> Dear Reviewer rKAZ,
>
>
> We sincerely thank you for your thoughtful and encouraging review. We are grateful that you recognize the novelty and significance of our work. We apologize for the delayed response, which was due to the time required to run additional image generation experiments.
>
>
> Below, we provide detailed point-by-point responses to address your questions and concerns.
>
> ---
> >`C1: First and foremost, I would've appreciated more experiments on challenging generative tasks. ... `
> ---
>
> Thank you for this valuable feedback and your appreciation of our theoretical contributions. We agree that demonstrating GFM on more diverse real-world tasks would strengthen the empirical validation. In response to your suggestion, we have conducted additional experiments on the following applications:
>
>
>
> 1. **Constrained time series generation**: We generate realistic traffic flow sequences subject to physical and statistical constraints [1,2], providing synthetic data for augmenting limited datasets in transportation systems. Using PEMS-BAY dataset [2], our experiments (Sec. 7.3) show:
>      - GFM achieves 100% constraint satisfaction versus 88.5% for vanilla flow matching.
>      - Projection methods increase inference time from 0.31s (FM) to 49.7s (160× slower), while GFM requires only 0.43s
>      - GFM preserves better distribution quality with KS statistic compared to projection methods
>
> 2. **Constrained image generation**: We generate watermarked images with polytope constraints over designated pixels, critical for AI content provenance and copyright protection. Fine-tuning from CIFAR-10 flow-matching checkpoints [4], our results (Sec. 7.4) show:
>    - GFM achieves 100\% feasibility, while vanilla flow matching only reaches 76\%.
>    - GFM attains better FID scores than reflection and projection methods.
>    - GFM's training and inference times match vanilla methods, whereas projection and reflection incur significant computational overhead
>    - Generated images of GFM are visually comparable to unconstrained baselines.
>
> **Note on experimental setup**: Due to limited time and GPU resources during rebuttal, we fine-tuned all baselines (in 200 epoch) from the checkpoint. Thus, the reported FID scores are higher than SOTA results for unconstrained FM models [4]. We will continue training until full convergence for the final manuscript.
>
> In the revised manuscript, our experiments across five diverse tasks demonstrate GFM's broad effectiveness:
>    - complex 2D constraints (Sec. 7.1)
>    - robotic configuration generation (Sec. 7.2)
>    - combinatorial optimization problems (Sec. 7.5)
>    - (New) constrained time series generation (Sec. 7.3)
>    - (New) watermarked image generation (Sec. 7.4)
>
> These span high-dimensional images, temporal sequences, and combinatorial domains with complex convex constraints, directly addressing concerns about practical applicability and real-world relevance.
>
> ---
>
> - [1] Narasimhan et al., Constrained Posterior Sampling: Time Series Generation with Hard Constraints. arXiv 2024
> - [2] Li et al., Diffusion convolutional recurrent neural network: Data-driven traffic forecasting. ICLR 2018
> - [3] Liu et al., Mirror diffusion models for constrained and watermarked generation. NeurIPS 2023
> - [4] Tong et al., Improving and generalizing flow-based generative models with minibatch optimal transport. TMLR 2024
>
>
>
> ---
> >`C2.1: I think Figure 1 captures the core idea of your work very well. Why don't you move it to an earlier page? `
> ---
>
> Thank you for this excellent suggestion. We agree that Figure 1 effectively captures the core conceptual framework of our method. We have moved it to page 1 in the revised manuscript to provide readers with an intuitive overview before diving into technical details.
>
>
>
>
> ---
> >`C2.2: Also, figure captions in general could be more elaborate and contain more details.`
> ---
>
> We appreciate this feedback. We have revised figure captions to be more informative.
>
>
>
> ___
> >`C2.3: Excessive use of bold faces and color highlights hurts readability in my opinion.`
> ___
>
> Thank you for pointing this out. We have significantly reduced the use of bold text and color highlights throughout the manuscript to improve readability. We now use these formatting elements sparingly and only to emphasize critical technical terms or key results.

---

### Official Review · Reviewer_B79x · 2025-10-30

**Soundness:** 2
**Presentation:** 4
**Contribution:** 2
**Rating:** 2
**Confidence:** 3

**Summary:**

The paper introduces Gauge Flow Matching, a constrained sampling approach that generalizes to arbitrary convex sets. The methodology builds on prior work for mirror mappings and reflected diffusion, originally proposed within diffusion frameworks, providing a generalization for flow matching with broader coverage of convex constraints. The experimental analysis on synthetic settings reports promising results for improving fidelity to the training distribution while improving runtime over existing constrained sampling approaches.

**Strengths:**

- **Motivation and Articulation:** The paper's motivation is well established and articulated. The writing is fairly easy to follow, and the methodology is communicated well.

- **Theoretical Support:** The theoretical grounding presented in the paper strengthens the claims made by the authors. The analysis appears to be sound, and the provided bounds contribute to the soundness of the paper.

- **Run-time Efficiency:** The efficiency arguments are an interesting contribution. While I believe the authors could make a better case for this by motivating it with time-sensitive, real-world experimental settings, I believe there are some relevant applications where this could be applied.

**Weaknesses:**

- **Limited Contribution:** The novelty of the method seems to be directly tied to the application of the existing gauge mapping [1] within flow matching models. The actual integration of this into the flow matching model is very closely tied to prior work on mirror maps and reflection based techniques. While the authors do point out that mirror maps do not generalize to arbitrary convex sets, it seems that beyond this mapping "substitution", the contribution is fairly limited.

- **Limited Empirical Evaluation:** While the experimental evaluation provides strong results, it relies solely on synthetic test cases with little to no real-world meaning. Furthermore, for many of these test cases, rejection sampling seems to be the better approach, as feasibility rates are higher than 90%, and samples have higher fidelity to the training set. Hence, the results would be much more compelling if they were applied to real-world settings where constrained sampling would be practically applied; this seems to be a missed opportunity, especially if such settings could provide a compelling case for the introduced efficiency.

- **Selected Baselines:** It is surprising that the authors have not chosen to compare to mirror diffusion models [2] in any of their analysis. Given how closely the method overlaps with mirror mappings, the omission of this comparison makes it difficult to assess whether the gauge map is effective -- which is the core contribution of the work. Furthermore, this would be a valuable comparison to assess how different bijective mappings influence the generation quality.

---

[1] Tabas, Daniel, and Baosen Zhang. "Computationally efficient safe reinforcement learning for power systems." 2022 American Control Conference (ACC). IEEE, 2022.

[2] Liu, Guan-Horng, et al. "Mirror diffusion models for constrained and watermarked generation." Advances in Neural Information Processing Systems 36 (2023): 42898-42917.

**Questions:**

- In Tables 3 and 5, the training time and inference times are lower than the "vanilla" models. Can the authors explain why this would be?

- Along similar lines as the previous question, the runtime for the reflection and projection algorithms (Tables 3 and 4) seem unexpectedly high. Can the authors speak to how these were implemented? Given the simple nature of these constraints, it would seem these operations could be more efficiently computed (e.g., an efficient closed-form projection operator).

---

> ### Author Response · Authors · 2025-11-27
> **Response (Part I)**
>
> Dear reviewer B79x,
>
> We sincerely thank you for acknowledging our theoretical contributions and the computational efficiency of our framework. We apologize for the delayed response, which was due to the time required to run additional image generation experiments.
>
> We appreciate your constructive feedback and provide point-by-point responses below to address your concerns.
>
>
>
> ---
> >`C1: Limited Contribution: The novelty of the method seems to be directly tied to the application of the existing gauge mapping [1] within flow matching models. The actual integration of this into the flow matching model is very closely tied to prior work on mirror maps and reflection-based techniques. While the authors do point out that mirror maps do not generalize to arbitrary convex sets, it seems that beyond this mapping "substitution", the contribution is fairly limited.`
> ---
>
> We appreciate your feedback and respectfully clarify our contributions, which extend **fundamentally** beyond combining existing techniques:
>
>
> - **Generalized gauge mapping (Remark 1, p. 4)**: Reference [1] only establishes closed-form homeomorphisms between unit balls and polytopes. We develop a complete framework for gauge mappings between arbitrary convex sets, including closed-form solutions and efficient bisection algorithms for general cases. This non-trivial generalization enables applications far beyond [1].
>
> - **Bi-Lipschitz characterization (Prop. 4.1)**: We derive explicit Bi-Lipschitz bounds for gauge mappings, essential for establishing data regularity preservation (Prop. 5.1) and bounded distribution error (Prop. 5.2). These quantitative bounds provide rigorous theoretical guarantees absent in prior work on gauge mappings [1] and mirror maps [2].
> - **Extension to Non-convex Sets (Sec. 6)**: We extend gauge map construction to star-convex and geodesic-convex sets, with experiments in Sec. 7.1 Demonstrating generality and performance. While previous gauge mappings [1], mirror maps [2], and reflection [3] do not directly extend to those sets.
> - **Fundamental advantages over mirror map (Remark 2, p. 5)s**: Mirror maps only apply to specific sets (balls, simplices, orthogonal linear constraints) with closed-form expressions, and exhibit boundary singularities (mapping boundary points to infinity), precluding distribution approximation guarantees. Our gauge mapping maintains bounded distortion at boundaries and provides rigorous error guarantees for arbitrary convex sets (Prop. 5.2).
> - **Fundamental advantages over direct refection (Remark 3, p. 6)**:  Reflection methods are computationally expensive beyond simple sets and require a prior distribution within the target constraint set C, which is challenging to sample from over general convex sets. Our framework transforms data over a unit ball (easily sampled) and implements closed-form reflection with $O(n)$ complexity.
>
>
> Thus, our framework fundamentally enables bijective mapping-based generative modeling for arbitrary convex sets with rigorous theoretical guarantees and computational efficiency. This represents a non-trivial generalization rather than a simple substitution of existing techniques.
>
>
> - [1] Tabas and Zhang. Computationally efficient safe reinforcement learning for power systems. ACC 2022.
> - [2] Liu et al. Mirror diffusion models for constrained and watermarked generation. NeurIPS 2023.
> - [3] Xie et al., Reflected Flow Matching. ICML 2024

---

> ### Author Response · Authors · 2025-11-27
> **Response (Part II)**
>
> ---
> >`C2: Limited Empirical Evaluation:  While the experimental evaluation provides strong results, it relies solely on synthetic test cases with little to no real-world meaning. Furthermore, for many of these test cases, rejection sampling seems to be the better approach, as feasibility rates are higher than 90\%, and samples have higher fidelity to the training set. Hence, the results would be much more compelling if they were applied to real-world settings where constrained sampling would be practically applied. this seems to be a missed opportunity, especially if such settings could provide a compelling case for the introduced efficiency`
> ---
>
> We appreciate your feedback and provide the following clarifications and additional experiments:
>
> **Metropolis (Rejection) sampling limitations**: While rejection sampling shows competitive performance on low-dimensional examples with high feasibility rates (>90\%), it exhibits critical drawbacks: (1) higher MMD scores, indicating worse distribution matching quality, (2) longer inference times due to repeated sampling and rejection, (3) prior feasible samples from the target constraint set are expensive to sample.
>
> These limitations become prohibitive in high-dimensional settings where the volume of feasible regions decreases exponentially (i.e., *"curse of dimensionality"*), making rejection sampling inefficient. We provide two illustrative examples:
>   - Low-dimensional case (joint linear and quadratic convex set): Rejection sampling requires 6 seconds (10× longer than GFM) to generate 1,000 samples and achieves an MMD of 0.13, whereas GFM achieves an MMD of 0.0035 with significantly faster inference.
>   - High-dimensional case (e.g., relaxed combinatorial sampling): The initial prior sampling for rejection methods incurs computational complexity of $O(n^8\log n)$, making both training and inference prohibitively expensive.
>
>
> **Real-world applications**: To address your concern about practical applicability, we have added the following real-world experiments:
>
>
>
> 1. **Constrained time series generation**: We generate realistic traffic flow sequences subject to physical and statistical constraints [1,2], providing synthetic data for augmenting limited datasets in transportation systems. Using PEMS-BAY dataset [2], our experiments (Sec. 7.3) show:
>      - GFM achieves 100% constraint satisfaction versus 88.5% for vanilla flow matching.
>      - Projection methods increase inference time from 0.31s (FM) to 49.7s (160× slower), while GFM requires only 0.43s
>      - GFM preserves better distribution quality with KS statistic compared to projection methods
>
> 2. **Constrained image generation**: We generate watermarked images with polytope constraints over designated pixels, critical for AI content provenance and copyright protection. Fine-tuning from CIFAR-10 flow-matching checkpoints [4], our results (Sec. 7.4) show:
>    - GFM achieves 100\% feasibility, while vanilla flow matching only reaches 76\%.
>    - GFM attains better FID scores than reflection and projection methods.
>    - GFM's training and inference times match vanilla methods, whereas projection and reflection incur significant computational overhead
>    - Generated images of GFM are visually comparable to unconstrained baselines.
>
> **Note on experimental setup**: Due to limited time and GPU resources during rebuttal, we fine-tuned all baselines (in 200 epochs) from the checkpoint. Thus, the reported FID scores are higher than SOTA results for unconstrained FM models [4]. We will continue training until full convergence for the final manuscript.
>
> In the revised manuscript, our experiments across five diverse tasks demonstrate GFM's broad effectiveness:
>    - complex 2D constraints (Sec. 7.1)
>    - robotic configuration generation (Sec. 7.2)
>    - combinatorial optimization problems (Sec. 7.5)
>    - (New) constrained time series generation (Sec. 7.3)
>    - (New) watermarked image generation (Sec. 7.4)
>
> These span high-dimensional images, temporal sequences, and combinatorial domains with complex convex constraints, directly addressing concerns about practical applicability and real-world relevance.
>
> ---
>
> - [1] Narasimhan et al., Constrained Posterior Sampling: Time Series Generation with Hard Constraints. arXiv 2024
> - [2] Li et al., Diffusion convolutional recurrent neural network: Data-driven traffic forecasting. ICLR 2018
> - [3] Liu et al., Mirror diffusion models for constrained and watermarked generation. NeurIPS 2023
> - [4] Tong et al., Improving and generalizing flow-based generative models with minibatch optimal transport. TMLR 2024

---

> ### Author Response · Authors · 2025-11-27
> **Response (Part III)**
>
> ---
> >`C3: Selected Baselines: It is surprising that the authors have not chosen to compare to mirror diffusion models [2] in any of their analysis. Given how closely the method overlaps with mirror mappings, the omission of this comparison makes it difficult to assess whether the gauge map is effective -- which is the core contribution of the work. Furthermore, this would be a valuable comparison to assess how different bijective mappings influence the generation quality.`
> ---
>
> We appreciate this suggestion and provide the following clarification:
>
> - **Comparison of mirror maps**: We are aware of the mirror diffusion model, and we have extensively discussed it in our works (Related work line 94-100, Remark 2 line 252-259).
>   - However, we did not include them as experimental baselines because mirror maps only admit closed-form solutions for specific constraint sets (e.g., simplices and balls), whereas our experiments primarily focus on complex convex sets (e.g., PSD cones) where mirror maps are not directly applicable. This fundamental limitation—the core motivation for our work—makes direct experimental comparison infeasible for most of our test cases.
>
>
> - **Head-to-head comparison**: To address your concern, we have added a direct comparison to mirror-map generative models on simplex constraints (Appendix E.5, page 29), where both methods have closed-form solutions. Our results show that:
>   - GFM achieves superior generation quality (lower MMD): 6.5 vs. 8.3 for 2D simplex, and 2.5 vs. 6.0 for 50D simplex.
>   - Mirror maps are slightly faster in training and inference due to lower computational complexity in the unconstrained dual space.
>
> This comparison demonstrates that our gauge mapping provides competitive performance on simple cases where mirror maps apply, while our GFM model uniquely enables generative modeling on arbitrary convex sets.

---

> ### Author Response · Authors · 2025-11-27
> **Response (Part IV)**
>
> ---
> >`Q1: In Tables 3 and 5, the training time and inference times are lower than the "vanilla" models. Can the authors explain why this would be?`
> ---
>
> Thank you for catching this. After reviewing our results, we identified the issue: our GFM method shows slightly longer or comparable times to FM (expected due to gauge calculations), but vanilla DDPM shows anomalously long times in Tables 3 and 5.
>
> After verifying our code implementation, we believe the anomalous results were caused by server load fluctuations. The original experiments were conducted sequentially on a shared server before the conference deadline period, which introduced large temporal variations in timing measurements that disproportionately affected certain runs.
>
> To ensure accurate measurements, we have re-run all inference timing and epoch training time comparisons on an unoccupied server.  Updated results for Tables 3 and 5 (as well as other tables in the main paper):
>   - GFM requires slightly longer computation time than vanilla FM and DDPM
>   - This increase is expected and attributable to gauge calculations for inverse/forward transformations
>
>
> We appreciate your careful attention to this detail. We have updated Tables 3 and 5 with these corrected measurements and will ensure the revised manuscript presents all timing comparisons accurately and consistently.
>
>
>
>
> ___
> `Q2: Along similar lines as the previous question, the runtime for the reflection and projection algorithms (Tables 3 and 4) seems unexpectedly high. Can the authors speak to how these were implemented? Given the simple nature of these constraints, it would seem these operations could be more efficiently computed (e.g., an efficient closed-form projection operator).`
> ___
>
> Thank you for this question about the computational costs of baseline methods. We clarify their implementation and explain the inherent complexity that leads to the observed runtimes.
>
> 1. **Reflection**:
> We implement reflection following the closed-form expressions detailed in Table 9 and Alg. 4 (Appendix E.2, page 28). This requires computing normal vectors at constraint boundaries. While simple linear and quadratic constraints admit efficient closed-form calculations, the computational bottleneck arises from:
>
>    - **Multiple constraint handling**: For constraints defined by intersections of multiple sets (e.g., polytopes intersecting with ellipsoids), the reflection method could perform iterative reflections near multiple constraint boundaries at every integration step for some samples. This per-step cost accumulates significantly over the entire sampling trajectory.
>
>    - **Complex geometries**: For star-convex sets and other complex constraints, computing reflection vectors requires calculating normal vectors via automatic differentiation (torch.autograd), which incurs substantial overhead at the reflection step.
>
>
> 2. **Projection**:
> We emphasize that closed-form projection operators generally do not exist for the complex constraints in our experiments, such as intersections of polytopes and ellipsoids, and star-convex constraints.
>
>    - Therefore, we employ state-of-the-art optimization solvers: (i) **MOSEK** for convex projection and (ii) **IPOPT** for non-convex cases.
>
>    - While these are highly optimized solvers, iteratively solving a constrained optimization problem at **each** sampling step for **each** sample (whenever the ODE trajectory violates constraints) remains computationally expensive. This cost is particularly significant for high-dimensional problems or tight constraint sets where violations occur frequently throughout the sampling process.
>
> In contrast, our gauge transformation approach mainly incurs computational cost at the final decoding step, where we apply closed-form transformations or use bisection (for general constraints). This fundamental difference explains why our method achieves superior runtime when handling complex constraints strictly throughout generation.

---

### Official Review · Reviewer_FHKh · 2025-10-31

**Soundness:** 3
**Presentation:** 3
**Contribution:** 3
**Rating:** 6
**Confidence:** 4

**Summary:**

GFM enforces hard constraints by bijective mapping any compact convex support set  "C"   to the unit ball B via a gauge map,training and generating inside B (with cheap boundary reflection) and mapping back, so samples are strictly feasible by construction. The gauge map is bi-Lipschitz, keeping distortion bounded and preserving distributional regularity when transferring training/sampling between "C"  and B. The framework extends to star-convex and geodesic-convex sets via appropriate generalized gauge maps. Experiments show 0% violations, strong distributional fit (e.g., low MMD), and faster generation than prior constrained methods.

**Strengths:**

1. The author proposes a generalized gauge mapping for constrained generation on any compact convex.The map Φ:B^2↔C is bijective and bi-Lipschitz with explicit bounds, which yields a clean distributional error transfer for any p-Wasserstein distance(Proposition 4.1),so training in B_2 preserves accuracy up to a bounded factor; feasibility is strict by construction.Extensions to star-convex and geodesic-convex sets (Appendix A) further broaden scope.

2. Computational efficiency:overall cost ≈ unconstrained models + small mapping overhead.Gauge values are closed-form for many sets (linear/quadratic/Second Order Cone/...); for general convex sets a 1D bisection along rays computes boundary intersections rapidly. The approach scales to high-D and complex constraints.

3. The empirical evaluation is convincing: 0% violations, low MMD (often outperforming projection/reflection/Metropolis baselines), and faster inference —showing constraints do not degrade sample quality.

4. The paper is clearly written:clear positioning (comparative table), intuitive figures for the gauge map, and precise propositions.

**Weaknesses:**

1. Sensitivity to interior point & data assumptions:Robustness still hinges on a good interior point, which can be non-trivial in high dimensions; moreover, requirement of the data distribution may limit real-world performance.

2. Although many cases admit closed-form or 1D bisection, per-sample gauge evaluation could dominate for very high-D or costly oracles (large LPs/implicit constraints),as shown in Figure 6.

3. Reproducibility:Many evaluations use programmatically generated target distributions tied to the constraints. While standard for this setting, the paper would be stronger with public tasks + released seeds/code, plus tests on real constrained domains (e.g., PSD cones, Birkhoff polytope, trajectory constraints).

**Questions:**

1. How do the authors envision extending GFM to more general non-convex domains beyond the star-convex and geodesic-convex cases?

2. The paper mentions that extending GFM to discrete generation via relaxation or embedding is a promising direction. Could the authors elaborate on how this might be achieved?

---

> ### Author Response · Authors · 2025-11-27
> **Response (Part I)**
>
> Dear Reviewer FHKh，
>
> Thank you for your positive comments and acknowledgment of the theoretical contributions and computational efficiency of our framework.
> We apologize for the delayed response, which was due to the time required to run additional image generation experiments.
> We provide point-by-point responses below to address your concerns.
>
> ---
> >`C1.1: Sensitivity to interior point: Robustness still hinges on a good interior point, which can be non-trivial in high dimensions`
> ---
>
> We appreciate your concern regarding the selection of interior points. This issue has been thoroughly addressed below Prop. 4.1, with additional sensitivity analysis in the Appendix (Figure 14, page 35). We clarify:
>
> -  **Explicit optimization of interior points**: The interior point directly affects the Lipschitz constant of the gauge mapping, which determines the distribution error bound (Prop. 5.2). We address this by:
>      - Deriving an explicit Lipschitz bound depending on the interior point (Prop. 4.1), enabling systematic optimization by minimizing this constant, which is equivalent to finding a well-centered point.
>      - We compute the central interior point via convex optimization (minimizing constraint residuals), solvable in polynomial time using standard solvers (e.g., MOSEK). This computation is performed **once** before training with negligible overhead to the overall training pipeline.
>        - e.g., training the model in Sec. 7.5 needs about 900 seconds in total 1000 epochs for a 100x100 dimension problem, while calculating an interior point before training needs less than 5 seconds.
>
> -  **Empirical evidence**: Our experiments demonstrate effectiveness across high-dimensional datasets using the computed central interior point. Sensitivity analysis (Figure 14) confirms that our selection strategy maintains robust performance even with suboptimal points.
>
> Therefore, by explicitly optimizing interior point selection through our principled approach, we effectively ensure robustness in high-dimensional settings.
>
>
>
> ---
> >`C1.2: data assumptions: requirement of the data distribution may limit real-world performance`
> ---
>
> We appreciate this concern and address it from both theoretical and practical perspectives:
>
>
> - **Standard assumptions in the literature**: The regularity requirements are standard for flow matching theoretical analysis, not specific to our method. Our gauge mapping provably preserves these properties (Prop. 5.1). We adopt assumptions from recent works [1], which are already minimal (e.g., non-vanishing density) compared to earlier works requiring covariance conditions [2], κ-semi-log-concavity [3], or convex support [4]. See Appendix D.1 (p. 25) for detailed discussion.
>
> - **Practical performance**: In practice, generative models operate on data samples without explicitly verifying theoretical assumptions. Our method similarly generates valid samples across diverse problems without assumption verification. Experimental results demonstrate practical robustness beyond theoretical requirements, indicating these assumptions enable worst-case guarantees rather than limit real-world applicability.
>
> ---
>
> - [1] Wan et al., Elucidating flow matching ODE dynamics with respect to data geometries. ICML 2024.
> - [2] Benton et al., Error bounds for flow matching methods. TMLR. 2023
> - [3]Gao et al., Gaussian interpolation flows. JMLR 2024
> - [4] Xie et al., Reflected Flow Matching. ICML 2024

---

> ### Author Response · Authors · 2025-11-27
> **Response (Part II)**
>
> ---
> >`C2: ..., per-sample gauge evaluation could dominate for very high-D or costly oracles (large LPs/implicit constraints), ...`
> ---
>
> We appreciate this concern and clarify the efficiency of gauge evaluation from both empirical and theoretical perspectives.
>
> **Negligible cost in the generation pipeline:** Gauge evaluation occurs only in the final transformation step, adding negligible overhead compared to forward integration (e.g., 100-step Euler). Figure 6 (in the original manuscript) exactly demonstrates its high efficiency, e.g.,
>   - **Batch computation**: For 10,000 samples in 500-dim space, gauge computation takes ~1s for linear constraints.
>   - **High dimensions**: In 3,000-dim space, gauge computation for 256 samples takes 0.5s across various convex constraints
>   - **Many constraints**: For 512 constraints in 500-dim space, computation takes 0.02s (linear) to 6s (other convex constraints)
>   - **Complex polynomials**: For polynomial constraints with 1.7 million terms, bisection computation for 1,000 samples takes 0.492s
>
> From a theoretical perspective, gauge function evaluation only requires a **membership oracle** (verifying feasibility), which is substantially cheaper than alternatives [1]:
> - **Projection** requires a *quadratic optimization oracle*, necessitating solving constrained optimization problems.
> - **Reflection** requires a *separation oracle*, needing computation of hyperplanes at constraint boundaries.
>
> For **Implicit constraint** (e.g., black-box simulators), the membership oracles remain efficient compared to methods in constrained generation literature, e.g.,
>   - The projection-guided generation [1] needs a gradient (via random perturbation) from the implicit constraint at **each** integration step, necessitating constraint evaluation for a set of samples.
>   - Our bisection-based gauge calculation at the **final** step requires only $O(\log(1/\epsilon))$ evaluations for the implicit constraint (if it is convex).
>
> Thus, our  GFM achieves superior efficiency compared to existing constrained generation methods, both theoretically (oracle complexity) and empirically (wall-clock time).
>
> - [1] Mhammedi Z. Efficient projection-free online convex optimization with membership oracle. COLT 2022
> - [2] Zampini et al., Training-free constrained generation with stable diffusion models. NeurIPS 2025.
>
>
> ---
> >`C3: Reproducibility: ... While standard for this setting, the paper would be stronger with public tasks + released seeds/code, plus tests on real constrained domains.`
> ---
>
> We thank the reviewer for this valuable suggestion.
>
> We provide detailed data generation procedures and training parameters in Appendix E, and commit to releasing complete code upon acceptance to ensure full reproducibility. To strengthen our paper via public tasks, we have included two new experiments:
>
> 1. **Constrained time series generation**: We generate realistic traffic flow sequences subject to physical and statistical constraints [1,2], important for data augmentation. Using the PEMS-BAY dataset [2], our experiments (Sec. 7.3) show:
>      - GFM achieves 100\% feasibility versus 88.5% for vanilla FM
>      - Projection increase inference time from 0.31s (FM) to 49.7s (160× slower), while GFM requires only 0.43s
>      - GFM preserves better distribution quality compared to Projection.
>
> 2. **Constrained image generation**: We generate watermarked images with polytope constraints over designated pixels, critical for AI content provenance [3]. Our results (Sec. 7.4) show:
>    - GFM achieves 100\% feasibility, while vanilla flow matching only reaches 76\%.
>    - GFM attains better FID scores than Reflection and Projection.
>    - GFM's training and inference times match vanilla FM.
>
> **Note on experimental setup**: Due to limited time and GPU resources during rebuttal, we fine-tuned all baselines (in 200 epochs) from the CIFAR-10 FM checkpoints [4]. Thus, the reported FID scores are higher than SOTA results for unconstrained FM models [4]. We will continue training until full convergence in the final manuscript.
>
>
> In the revised manuscript, we include five diverse tasks,
>    - complex 2D constraints (Sec. 7.1)
>    - robotic configuration generation (Sec. 7.2)
>    - combinatorial optimization problems (Sec. 7.5)
>    - (New) constrained time series generation (Sec. 7.3)
>    - (New) watermarked image generation (Sec. 7.4)
>
>    These tasks span watermarked images, temporal sequences, and combinatorial problems, directly addressing concerns about practical applicability and real-world relevance.
>
> ---
>
> - [1] Narasimhan et al., Constrained Posterior Sampling: Time Series Generation with Hard Constraints. arXiv 2024
> - [2] Li et al., Diffusion convolutional recurrent neural network: Data-driven traffic forecasting. ICLR 2018
> - [3] Liu et al., Mirror diffusion models for constrained and watermarked generation. NeurIPS 2023
> - [4] Tong et al., Improving and generalizing flow-based generative models with minibatch optimal transport. TMLR 2024

---

> ### Author Response · Authors · 2025-11-27
> **Response (Part III)**
>
> ---
> >`Q1: How do the authors envision extending GFM to more general non-convex domains beyond the star-convex and geodesic-convex cases?`
> ---
>
> Thank you for your interest in extending our method to general non-convex domains.
>
> Our current method relies on the fundamental topological equivalence between the constraint sets (convex, star-convex, and geodesically-convex) and a simple unit ball. For general non-convex sets, such topological equivalence may not exist, which poses theoretical challenges.
>
> However, we envision a promising extension using set decomposition. Complex compact non-convex sets (e.g., disconnected sets) can be decomposed into a finite union of convex or geodesically convex subsets [1,2].
>  - We can then build conditional generative models for each subdomain and generate samples conditioned on the subdomain index, where subdomain index probabilities are directly estimated from data. This conditional decomposition framework is a classical approach in probabilistic modeling.
>  - Importantly, decomposing complex non-convex sets into multiple convex or geodesically-convex components has been effectively demonstrated in the robotics control domain [1,2].
>
> Therefore, we believe this represents a viable pathway to extend our approach to more general constraint sets, which we plan to explore in future work.
>
> ---
>
> - [1] Marcucci et al., Motion planning around obstacles with convex optimization. Science robotics. 2023.
> - [2] Cohn et al., Non-Euclidean motion planning with graphs of geodesically convex sets. The International Journal of Robotics Research. 2025.
>
> ---
> >`Q2: The paper mentions that extending GFM to discrete generation via relaxation or embedding is a promising direction. Could the authors elaborate on how this might be achieved?  `
> ---
>
> Thank you for your interest in extending our method to discrete domains.
>
> As briefly discussed in the conclusion section, the key idea is to leverage continuous relaxation techniques from combinatorial optimization [1] to embed discrete variables into continuous domains, perform flow matching in the relaxed space, and decode back to discrete structures.
>   - This approach has proven effective in recent work: such as Fisher Flow Matching [2], which embeds categorical variables onto spheres and uses greedy decoding for recover discrete structure, achieving superior efficiency compared to direct discrete modeling.
>
>   - Our gauge transformation framework further provides complementary tools for this paradigm, as many advanced relaxation methods (e.g., semi-definite relaxation, sum-of-square polynomial relaxation) induce convex constraint constraint in the relaxed space, which make our method can effectively handle.
>
> We believe this represents an exciting direction for future work, combining the efficiency of continuous relaxations with the theoretical guarantees our gauge transformation provides.
>
> ---
>
> - [1] Balas, lifting and extended formulation in integer and combinatorial optimization. Annals of Operations Research. 2005.
> - [2] Davis et al., Fisher flow matching for generative modeling over discrete data. NeurIPS 2024.

---

### Author Response · Authors · 2025-12-03
**General Response by Authors**

Dear Reviewers, ACs, and PCs,

We sincerely thank all reviewers for their constructive comments, and we extend our gratitude to the ACs and PCs for their dedicated efforts on our paper. We are deeply saddened by the unexpected accident that has affected our community, and we hope that everyone's efforts will be recognized and valued.


---

We first thank the reviewers for their acknowledgment of our work's originality, theoretical contributions, computational efficiency, and clarity of writing:

1. **Originality and significance**:
   - Reviewer rKAZ: "*` I find the idea in this paper sufficiently original and of significance, ... `*"
   - Reviewer FHKh: "*` The author proposes a generalized gauge mapping for constrained generation on any compact convex. ... Extensions to star-convex and geodesic-convex sets further broaden scope.`*"

2. **Theoretical contributions**:
    - Reviewer rKAZ: "*` The paper presents rigorous theoretical validation/analysis on the strict constraint guarantee ...`*"
    - Reviewer B79x:  "*` The analysis appears to be sound, and the provided bounds contribute to the soundness of the paper.`*"

3. **Computational efficiency**:
    - Reviewer FHKh: "*` The approach scales to high-D and complex constraints.`*"
    - Reviewer B79x: "*` The efficiency arguments are an interesting contribution.`*"

4. **Clear writing**:
    - Reviewer FHKh: "*` The paper is clearly written: ...`*"
    - Reviewer rKAZ: "*` I also appreciate the style and quality of writing.`*"

---


We have made the following revisions to directly address the reviewers' major concerns:

1.  **Empirical evaluation** on public and real-world benchmark tests → (Reviewers FHKh, B79x, rKAZ)
    - To address concerns regarding empirical evaluation on real-world tests with public benchmarks, we conducted additional experiments on:
        - *Constrained time-series generation* on the PEMS-BAY dataset (*Sec. 7.3*)
        - *Watermarked image generation* on the CIFAR-10 dataset (*Sec. 7.4*)
    - Both experiments further demonstrate the performance of our GFM framework, which achieved **better** generation quality than other constrained generation baselines (Reflection or Projection), while achieving significantly **lower** training and inference costs.
    - Together with three experiments in the original manuscript—complex 2D constraints (*Sec. 7.1*), robotic configuration (*Sec. 7.2*), and combinatorial problems (*Sec. 7.5*)—we believe these comprehensive empirical evaluations directly address the major comments on experimentation.


2. **Novelty and contribution** of our GFM framework → (Reviewer B79x)
    - We clarify that this has been extensively discussed in the original manuscript:
        - the originality of the proposed generalized gauge mapping (*Remark 1 \& Section 6*),
        - the theoretical contribution of the Bi-Lipschitz characterization (*Prop. 4.1*),
        - the fundamental theoretical and computational advantages over existing mirror maps and reflections (*Remarks 2 \& 3*).
    - We also note that the originality and significance of our proposed generalized gauge mapping have been **acknowledged** by the other two reviewers (see summary on **Originality and significance** above).
    - We believe these clarifications directly address Reviewer B79x's concerns regarding the novelty and originality of our generalized gauge mapping and the overall GFM framework.

---

We also address the remaining concerns of reviewers:

1. We clarify that interior point sensitivity is addressed through minimizing Lipschitz bounds (equivalent to central point identification), the distribution assumption is standard for theoretical analysis without precluding practical usage, and the superiority of gauge calculation efficiency both empirically and theoretically → (Reviewer FHKh).
2. We discuss extensions of our GFM framework to non-convex and discrete constraints → (Reviewer FHKh).
3. We conducted additional experiments over the simplex for direct comparison to mirror maps → (Reviewer B79x).
4. We clarify the cause of inconsistent inference times in Tables 3 \& 5, and updated the statistics after re-running experiments in a clean server → (Reviewer B79x).
5. We clarify that the high inference complexity of Reflection and Projection arises from their inefficiency in complex or multiple constraints, despite using industrial solvers → (Reviewer B79x).
6. We improved presentation through enhanced readability → (Reviewer rKAZ).
---

Through our comprehensive rebuttal, paper revisions, and additional extensive empirical evaluation, we believe we have thoroughly addressed all reviewer concerns.

Last, we highlight that our GFM framework is the **first** constrained generative model to provide *strict feasibility guarantees*, *computational efficiency*, and *distribution approximation bounds* for *general convex sets*, as well as *generalizations* to certain non-convex sets, representing a significant advancement in the field.

---

### Meta-Review · Area_Chair_TDsF · 2026-01-08

**Summary:**

Reviewers FHKh and rKAZ agree that the paper makes good contributions to the problem of constrained generative modeling by handing more general cases on the constraint set. Reviewer B79x thinks that the novelty is limited as it simply extends gauge mapping to flow matching models, and is closely tied to mirror maps and reflection based techniques. As noted in the rebuttal, the paper makes contributions on Generalized gauge mapping  and extensions to non-convex sets, which I believe is still a solid contribution.

Another concern that raised was regarding the lack of real-world benchmarks beyond synthetic data and the omission of key baselines like Mirror Diffusion Models (B79x). The authors provided some additional experiments on real world datasets in the rebuttal, which makes the paper more solid. While the results are not super strong compared to the baselines, they are reasonable better.

While the paper is borderline due to these concerns, I am inclined to accepting this paper as I believe the contribution makes nice combination of existing ideas (e.g., guage mapping + bijective mapping to simplify constrained generation), presents theoretical guarantees and complexity analysis which might be of interest to the community.

**Reviewer Concerns:**

The authors successfully addressed the empirical gap by adding two new real-world experiments (time-series and watermarked images) and provided a head-to-head comparison with Mirror Diffusion on some benchmarks. More experimental results on real-world benchmarks would have made the submission even stronger.

**Reviewer Scores:**

Reviewer FHKh would likely increase their score to a 7, as the authors provided the requested sensitivity analysis and demonstrated efficiency on high-dimensional image tasks. Reviewer B79x might increase the score, given that their primary criticisms regarding baselines, timing anomalies, and "synthetic-only" experiments were added. Reviewer rKAZ would maintain their 8, as the authors specifically implemented the image generation benchmarks that was requested.

---

### Decision · Program_Chairs · 2026-01-26

Accept (Poster)